

# Delocalized states in three-terminal superconductor-semiconductor nanowire devices

P. Yu[1], B. D. Woods[2,3], J. Chen[1], G. Badawy[4], E. P. A. M. Bakkers[4],
T. D. Stanescu[2] and Sergey M. Frolov[1*]

**1** University of Pittsburgh, Pittsburgh, PA 15260, USA
**2** Department of Physics and Astronomy, West Virginia University,
Morgantown, WV 26506, USA
**3** Department of Physics, University of Wisconsin-Madison, Madison, WI 53706, USA
**4** Eindhoven University of Technology, 5600 MB, Eindhoven, The Netherlands

* frolovsm@pitt.edu

## Abstract

We fabricate three-terminal hybrid devices consisting of a semiconductor nanowire segment proximitized by a grounded superconductor and having tunnel probe contacts on both sides. By performing simultaneous tunneling measurements, we identify delocalized states, which can be observed from both ends, and states localized near one of the tunnel barriers. The delocalized states can be traced from zero magnetic field to fields beyond 0.5 T. Within the regime that supports delocalized states, we search for correlated low-energy features consistent with the presence of Majorana zero modes. While both sides of the device exhibit ubiquitous low-energy features at high fields, no correlation is inferred. Simulations using a one-dimensional effective model suggest that the delocalized states, which extend throughout the whole system, have large characteristic wave vectors, while the lower momentum states expected to give rise to Majorana physics are localized by disorder. To avoid such localization and realize Majorana zero modes, disorder needs to be reduced significantly. We propose a method for estimating the disorder strength based on analyzing the level spacing between delocalized states.

The realization of Majorana zero modes (MZMs) has been proposed in an increasing number of materials and heterostructures [1–4]. Generated in pairs at the ends of a topological nanowire segment, MZMs are predicted to exhibit non-Abelian exchange statistics [5], which can be used as a resource for quantum computing [6,7]. Based on theoretical proposals [8,9], a number of experiments have been conducted to study the emergence of zero-bias conductance peaks (ZBCPs) at finite magnetic fields [10–14]. While the observation of ZBCPs is consistent with the presence of MZMs, it has been shown that topologically trivial states can also be at the origin of such observations [15–26]. In most cases, to identify non-Majorana states it is sufficient to analyze two-terminal measurements by testing the features associated

with tunneling into one end of the nanowire against specific Majorana signatures, such as stability with respect to local gate potentials. However, positive identification of MZMs will likely require three-terminal measurements, involving charge tunneling into both nanowire ends [25, 27–31]. More generally, the three-terminal technique, which is the focus of this work, is a powerful method for studying the localization of wavefunctions [32–34].

We fabricate three-terminal nanowire devices that nominally fulfill the basic requirements for Majorana physics, being built around InSb nanowires that have significant intrinsic spin-orbit coupling, with superconductivity induced by a NbTiN superconductor, and in the presence of a magnetic field that breaks time-reversal symmetry. Before applying the magnetic field, we observe a set of discrete states that simultaneously generate conductance signatures at both ends of a 400 nm proximitized segment. These states, which we dub "delocalized states" only appear above a certain S-gate voltage. By contrast, localized states visible only at one end of the device appear in a broader range of parameters. At finite magnetic field, we find ZBCPs on both sides of the device. However, they do not appear in a correlated manner, neither in the regime that supports delocalized states, nor outside that regime. We seek to shed light on these findings by performing numerical simulations based on an effective one-dimensional nanowire model with disorder. The modeling reveals a regime that supports delocalized states, as well as the presence of localized states. In the model, the delocalized states extend throughout the whole system and, consequently, are expected to generate experimental signatures at both ends of the system. These states have large quasi-momenta and energies above a threshold set by the disorder potential and are not non-local pairs of (localized) Majorana bound states or degenerate Andreev bound states. By contrast, the localized states have characteristic length scales shorter than the length of the system and are not expected to generate correlated experimental features. By comparing theory and experiment, we show that the three-terminal geometry enables one to estimate the localization length scales of low energy states within the nanowire device. In addition, we provide a rough estimate of the disorder level present in the nanowire by comparing the level spacing of delocalized states extracted from the experimental data to theoretical simulations.

Fig. 1(a) shows the scanning electron micrograph of the device A studied in this paper, and in one previous paper [25]. There are three contacts on the InSb nanowire: a NbTiN superconducting (S) contact in the middle and two Pd non-superconductor contacts $N_L$ and $N_R$ at the ends. We can apply voltage bias and measure current in different configurations. The red circuit shown in Fig. 1(a) is the most common configuration, where voltage bias is applied to S contact, and conductances $G_L$ and $G_R$ are measured at grounded $N_L$ and $N_R$ contacts. For measurement in Fig. 1(b), voltage bias is applied between $N_L$ and $N_R$ while S-contact is floating (black circuit in Fig. 1(a)). The device is electrostatically controlled by a 400 nm wide S-gate under the superconducting contact and the two tunneling barrier gates $T_L$ and $T_R$, both set to ensure depleted regions in the nanowire above them. The gates labeled $T_3$ are connected together and set so that the nanowire segments above them are highly electrostatically n-doped. More details about the fabrication and measurements can be found in the Methods section. Quantum states within the nanowire appear in transport measurements as peaks in conductance. We monitor where they are located within the nanowire by comparing the conductance signatures observed at the different terminals of our device. We present tunnel gate vs. S-gate scans of the zero bias conductance at zero magnetic field in Fig.1 (b)(c)(d). We classify the observed resonances into three groups. First, there are states localized on the right side, e.g., state R1, which shows dependence on the right tunnel gate $T_R$ and the S-gate (but is insensitive to $T_L$). Second, there are left localized states, e.g., L1, which exhibits no dependence on the right tunnel gate $T_R$ (but is sensitive to $T_L$, see supplementary information). The state $R_1$ only appears in right-side conductance $G_R$, while $L_1$ only appears in $G_L$. Both $R_1$ and $L_1$ are visible when current is passed from $N_L$ to $N_R$ (Fig.1 (b)). Third, apart from the

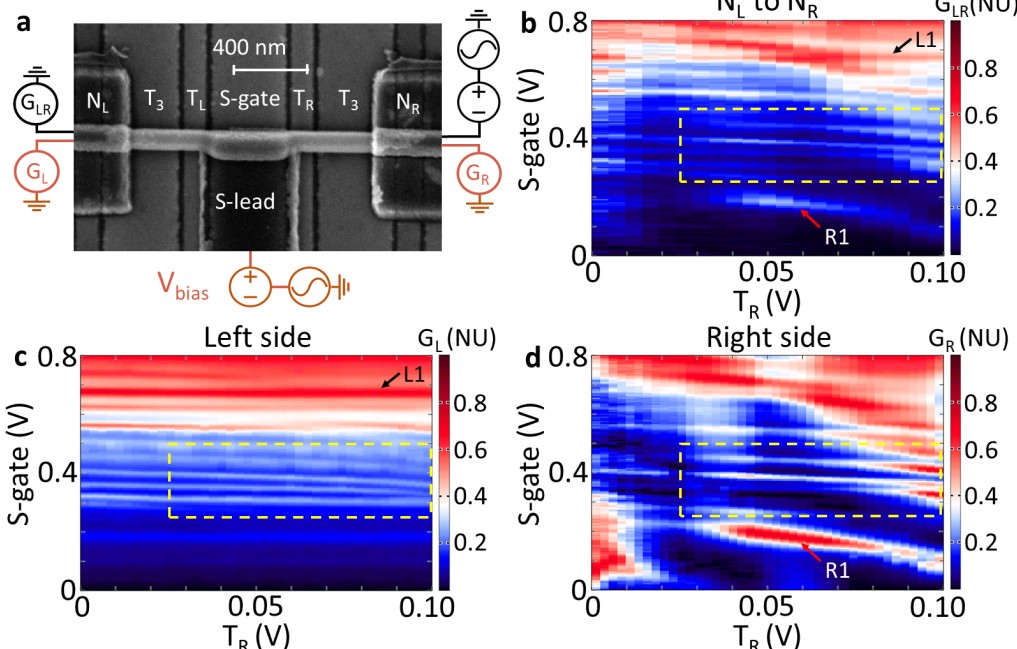

Figure 1: (a) Scanning electron micrograph of the measured device A, and the circuit diagrams for three-terminal measurements (red) and two-terminal $N_L$-$N_R$ measurements (black). (b) Zero-bias differential conductance measured between $N_L$ and $N_R$. (c) and (d) Differential conductance $G_L$ and $G_R$. Differential conductance in each row is in normalized units (NU) to make states more visible. Other gate voltages were $T_L$ = -0.17 V and $T_3$ = +2.64 V. Magnetic field is zero.

localized states, we also observe a sequence of states closely spaced in S-gate, within dashed rectangles in Figs. 1(b),(c),(d). These states appear in all three conductance measurements, and as we show in Fig. 2 they are sensitive to all gates. We call them delocalized and label them S1-S4 in subsequent figures.

It is worth noting that in this device localized states appear for both negative and positive S-gate voltages, while the delocalized states only appear for positive S-gate voltages. The full picture of state localization can be obtained by analyzing how $G_L$ and $G_R$ evolve with all three gates, $T_L$, $T_R$ and S-gate. In Fig. 2 we focus on the regime indicated by the dashed rectangle in Fig.1. The gate dependences of states S1-S4 are correlated between $G_L$ and $G_R$ [see Figs. 2(c),(f)]. In the $T_L$ scans we label S1-S4 based on their known positions from S-gate vs $T_R$ data [see Figs. 2(a) and (b)]. We notice that resonances strongly dependent on $T_L$, such as L0, exhibit anticrossings with S1-S4 that are reminiscent of double quantum dot charge stability diagrams (Figs. 2(d)(e)(f)). In this case the charging energy is reduced due to screening by the superconductor, so that these features are simply associated with wavefunctions having different spatial localization within the nanowire.

To conclude the zero magnetic field analysis, we show that the delocalized states S1-S4 can be probed from the two sides and are tunable with all gates. The presence of such states is expected, given that the nanowire segment between the tunnel barriers is 400 nm in length, only a factor of 4 greater than the nanowire diameter. What is more surprising is that within the same segment we observe wavefunctions significantly localized either near the left or the right end (L and R states).

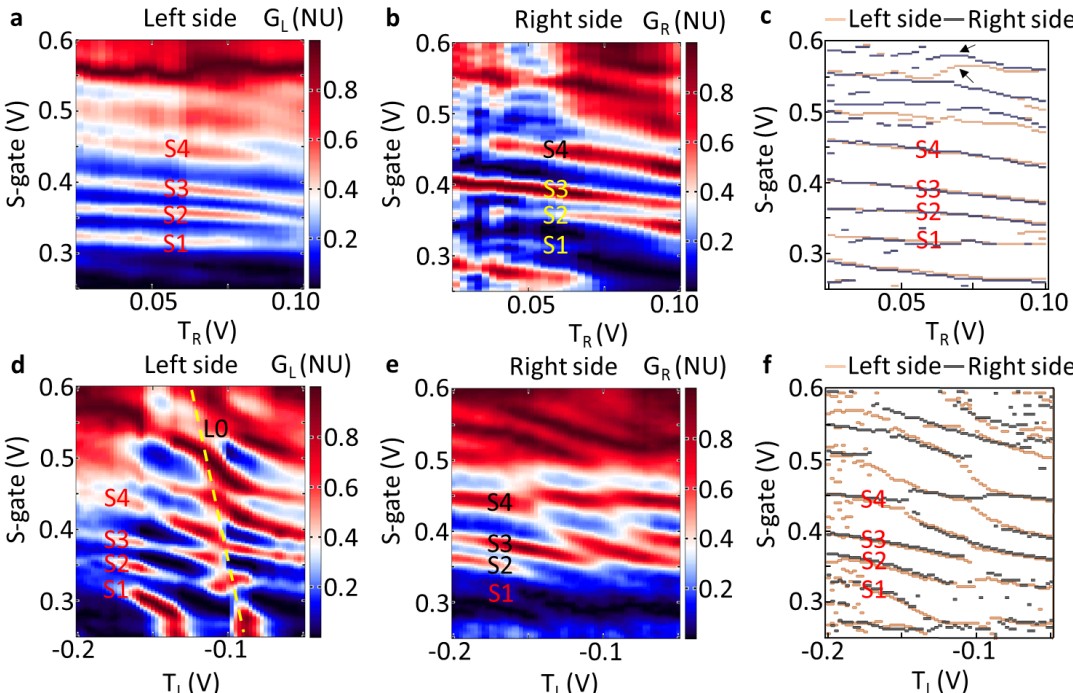

Figure 2: (a) and (b), Zero-bias differential conductances $G_L$ and $G_R$ as functions of $T_R$ and S-gate voltage focused on delocalized states S1-S4. Other gate voltages were $T_L = -0.17$ V and $T_3 = +2.64$ V. (c) Conductance maxima from panels (a) and (b). (d) and (e), $G_L$ and $G_R$ as functions of $T_L$ and S-gate voltage. Other gate voltages were $T_R = +0.075$ V and $T_3 = +2.64$ V. (f) Conductance maxima from panels (d) and (e). Differential conductance in each row is in normalized units (NU) to make states more visible. Magnetic field is zero.

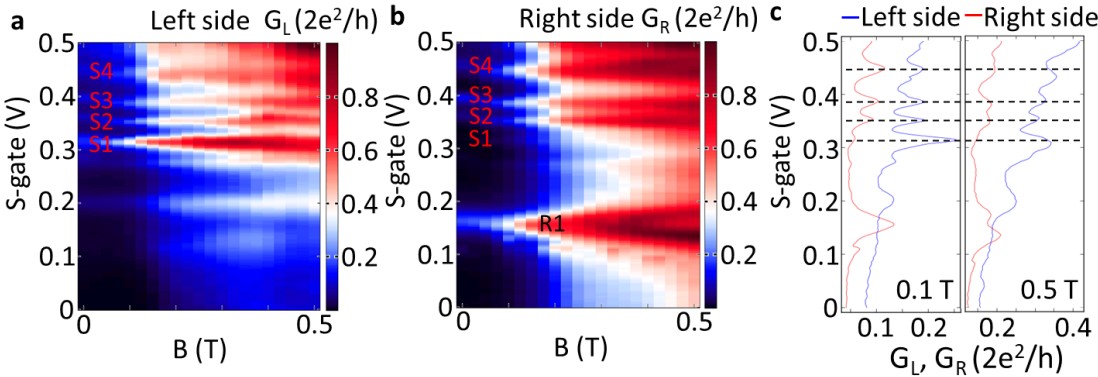

Figure 3: Magnetic field dependence of the localized and delocalized states. (a) and (b) Zero bias $G_L$ and $G_R$ measurements as functions of magnetic field B applied along the nanowire. Other gate voltages were $T_R = -0.17$ V, $T_R = +0.075$ V and $T_3 = +2.64$ V. (c) linecuts from (a) and (b). Dashed lines mark zero-field positions of states S1-S4.

In Fig. 3, we explore the magnetic field dependence of the resonances. States S1-S4 can be traced to finite field. However, the peak positions do not perfectly coincide in S-gate at B=0.5 T. The peaks S1-S4 broaden, presumably due to Zeeman effect. The Zeeman effect is more clear when the resonance R1 is examined. Generally, we find that some states dim,

while new resonances emerge at finite field and, as a result, states that appear correlated at zero magnetic field may not be so apparently correlated at finite magnetic field, which is the regime of interest for Majorana experiments.

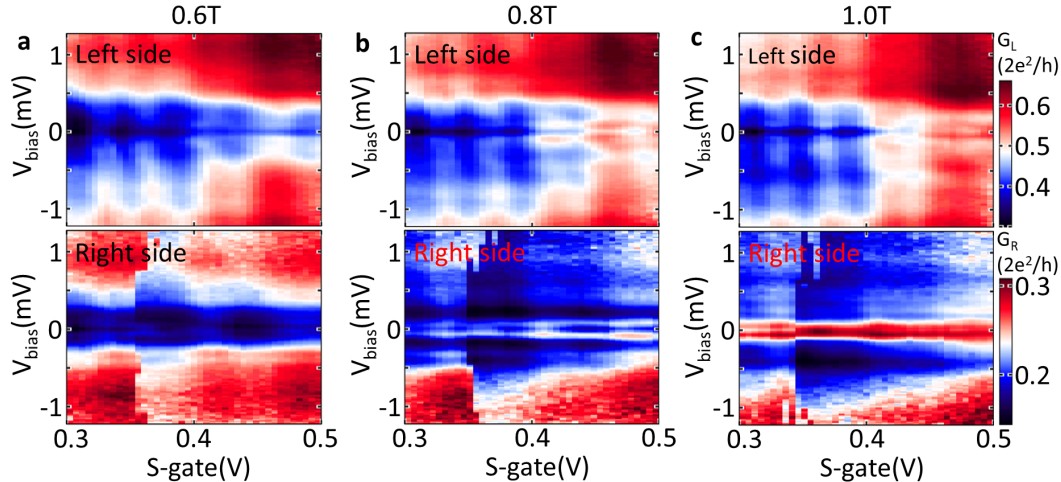

Figure 4: Low-energy states at finite magnetic fields. (a), (b) and (c) $G_L$ and $G_R$ at fields 0.6 T, 0.8 T and 1 T in the regime with delocalized states. Gates voltages were $T_R$ = -0.175 V, $T_R$ = +0.06 V and $T_3$ = +2.6 V.

Next, we search for Majorana signatures in this device. We previously demonstrated that for negative S-gate voltages it is possible to identify a zero-bias conductance peak partially consistent with Majorana physics from measurements of $G_L$ [25], but that peak was not accompanied by a correlated feature in $G_R$. Here we focus on the regime 0.3 V < S-gate < 0.5 V, where we observe the states S1-S4 delocalized across the nanowire. As shown in Fig. 4, while near-zero energy (low energy) states emerge on both sides, only the right side exhibits a clear zero-bias peak, with no correlated companion on the left side. In general, the merging and splitting of low-bias resonances is not correlated on both sides. At the highest field presented here, B = 1 T, it is also no longer possible to resolve resonances S1-S4 in both $G_L$ and $G_R$. We conclude that, while being able to observe delocalized states at zero (or low) magnetic field may be a necessary condition, it is clearly not a sufficient condition for observing Majorana signatures, such as correlated near-zero bias peaks, at finite magnetic field.

To get a better insight into the physics responsible for this phenomenology, we model the device using a one-dimensional effective model that incorporates information regarding the geometry of the system and assumes the presence of disorder. Note that the effective one-dimensional model is an appropriate approximation as long as the inter-subband spacing is larger than other relevant energy scales, e.g., the disorder strength. Inclusion of multiple subbands (with relatively small inter-subband gaps) is expected to enhance the disorder effects [20, 26]. As represented schematically in Fig. 5(e), the key components of the device included in the model are a 400 nm central region (blue) with proximity induced superconductivity and gate-controlled effective potential $V_S$ and two bare semiconductor segments (yellow) separated from the superconducting region by potential barriers (black) with heights $V_L$ (at the left end) and $V_R$ (right end). In addition, disorder is modeled as a position-dependent random potential with finite correlation length. Details regarding the model are provided elsewhere [20].

We calculate numerically the zero energy local density of states inside the left and right normal regions (Fig. 5). We assume a finite spectral broadening, $\eta = 30\ \mu$eV. Note that this yields a non-zero density of states at zero energy even though all states have a finite

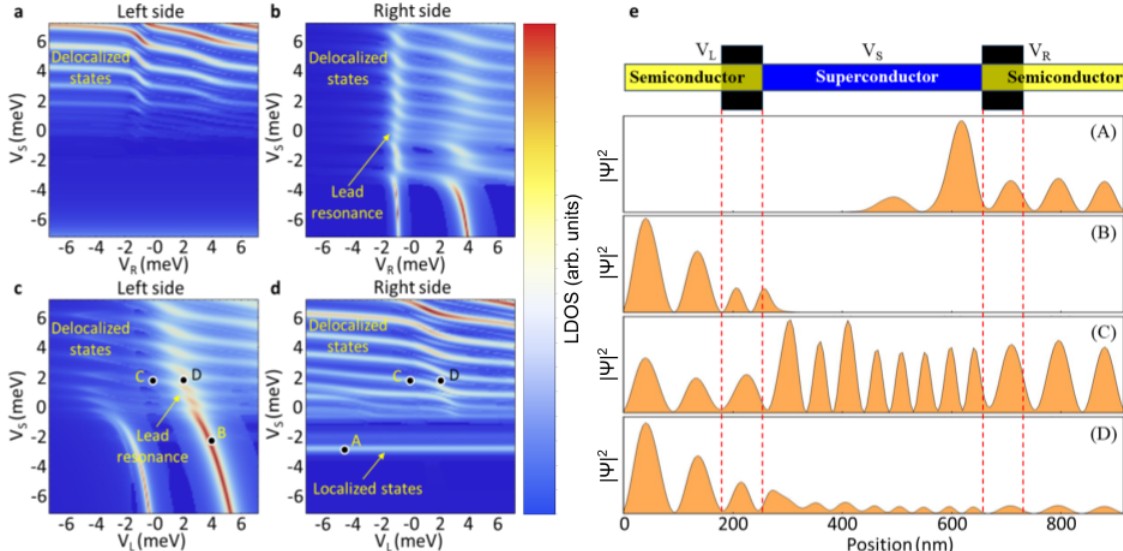

Figure 5: Numerically simulated color map of the zero energy local density of states inside the left and right normal regions as function of $V_S$, $V_L$ and $V_R$ (see schematic in (e)). Panels (a)(b) and (c)(d) correspond to $V_R = 0.5$ meV and $V_L = 0.5$ meV, respectively. (e) Position dependence of the wave function amplitudes corresponding to the lowest energy states associated with the points marked by letters (A-D) in panels (c), (d).

energy due to superconductivity. First, we notice the presence of the same key features that characterize the experimental results: i) delocalized states – features that appear on both sides of the system, are strongly dependent on $V_S$, and have a weaker dependence on $V_L$ and $V_R$; ii) localized states – features that only appear at one end and are practically independent of the barrier potential at the opposite end; iii) lead resonances – features that are strongly dependent on the barrier potential at the corresponding end and hybridize with the delocalized states. Changing the model parameters and the disorder realizations reveals that features (i) and (iii) are rather generic, while the presence of feature (ii), i.e., the localized states, requires a strong-enough disorder potential. The relative visibility of various features depends on the details of the model. For example, the length and height of the tunnel barriers controls the coupling between the superconductor and semiconductor regions, which in turn affects the visibility in the local density of states. Note, however, that the qualitative features of our results are unaffected by these details.

Having established the qualitative correspondence between the features observed experimentally and those obtained numerically, we determine explicitly the low-energy states associated with these features. The position dependence of the wave function amplitude $|\psi(x)|^2$ corresponding to the lowest energy states associated with the points marked by letters (A-D) in Figs. 5(a)-(d) is shown in Fig. 5(e). State (A) is, indeed, a localized state having a maximum near the right barrier and a "tail" that leaks into the right normal region. State (B), which is responsible for the lead resonance feature, is localized almost entirely inside the (left) normal region, with negligible weight inside the proximitized central segment. By contrast, state (C) is delocalized, extending throughout the whole system, which explains its visibility from both ends of the device. Finally, state (D), which is also associated with the lead resonance feature, is a hybrid state resulted from the hybridization of state (B), which is localized within the left normal region, with a delocalized state. An important point is that correlations associated with the presence of delocalized (type-C) states can only be observed above a certain value of the back gate potential, $V_S \approx 2$ meV, while the localized states (type-A) emerge at significantly

lower $V_S$ values. In this context, we note that in a system with multiple occupied bands, localized and delocalized features can coexist within the same range of gate potentials, $V_S$, but they will be associated with different subbands. More specifically, the localized features will be associated with the topmost occupied band, while the delocalized features will be generated by high-k states from the lower energy band(s). Applying a finite Zeeman field is expected to quickly reduce the energy of the low-k states (associated with the lower energy spin subband), while having a significantly weaker effect on the high-k, delocalized states due to a stronger spin-orbit coupling. Consequently, as the field increases, the zero energy local density of states will become dominated by localized (low-k) states, while the non-local correlations associated with the delocalized (finite energy) states will eventually be unobservable.

Next, by combining the insight provided by modeling with experimental information, we estimate the strength of disorder within the device. First, we note that states only become delocalized when their energy with respect to the bottom of the subband is comparable to or larger than the characteristic energy of the disorder. Intuitively, these high-energy states have a large quasi-momentum (and characteristic velocity), which makes them less susceptible to localization by disorder. By contrast, the low-energy states become localized in the presence of disorder and do not generate non-local features. Second, the level spacing increases (on average) with each new quantized state. Unfortunately, the exact relationship between the level spacing and the energy of the states is unknown, due to the random nature of the disorder potential. Nonetheless, we can calculate the statistical distribution of level spacing in the presence of disorder. Comparison of our experimental level spacing values with the level spacing statistics provides (probabilistic) information about the disorder strength in the system.

To determine the level spacing between delocalized states, we first estimate the lever arm of the S-gate by fitting the resonances of the bias vs. S-gate scan in Fig. 6(a) corresponding to the delocalized states. For weak semiconductor-superconductor coupling, the slope of these lines is simply equal to the lever arm $\alpha$. Our estimate is $\alpha \approx 62.5$ meV/V. The level spacing between the delocalized states is then the lever arm $\alpha$ times the distance $\Delta V_S$ between the resonances. We extract the following level spacing values: $\Delta E_{12} = 2.2$ meV, $\Delta E_{23} = 2.2$ meV, and $\Delta E_{34} = 3.4$ meV, yielding an averaged level spacing of $\Delta E_{ave} = 2.6$ meV.

To acquire statistics, we simulate an $L = 400$ nm nanowire segment in the presence of a random disorder potential $V$ with correlation function $\langle V(x)V(x+l) \rangle = U^2 \exp(-l^2/\xi^2)$, where $U$ is the disorder strength and $\xi = 15$ nm is the correlation length. The value of $\xi$ is chosen based on the results of Ref. [35] and assumes that disorder is mainly due to charge impurities. The effective mass is $m^* = 0.015 m_e$ and we neglect spin-orbit coupling. We define a state to be delocalized if its spectral weight within $L_{edge} = 100$ nm of each edge is over 10%. We then calculate the average level spacing $\Delta E_{ave}$ corresponding to the first four delocalized states. The results corresponding to different values of the disorder strength, $U$, are shown in Fig. 6 (b). For each value of $U$, we consider 2000 disorder realizations. The orange lines correspond to the median average level spacing, the boxes correspond to the 25-75% range, and the whiskers correspond to the 2nd and 98th percentiles. As expected, we find that the averaged level spacing $\Delta E_{ave}$ increases, on average, with increasing $U$. Notably, the spread around the average also increases with $U$. Assuming no a priori knowledge about the disorder level in our device, our experimental averaged level spacing of $\Delta E_{ave} = 2.6$ meV yields a distribution $f(U)$ of the disorder strength $U$, as shown in Fig. 6(c). We notice a peak in the distribution near $U \approx 5$ meV. Furthermore, we plot the cumulative distribution $F(U) = \int_0^U f(U') dU'$ in Fig. 6(d), which tells us the chances of having a disorder strength less than $U$, given our averaged level spacing $\Delta E_{ave}$. According to this result, there is about a 50% chance that $U < 5$ meV.

What does this tell us about possibly realizing MZMs in these devices? First, we emphasize that the Majorana physics expected to emerge at finite magnetic field is generated by low-energy states, at least at experimentally relevant intermediate fields. When nanowire states

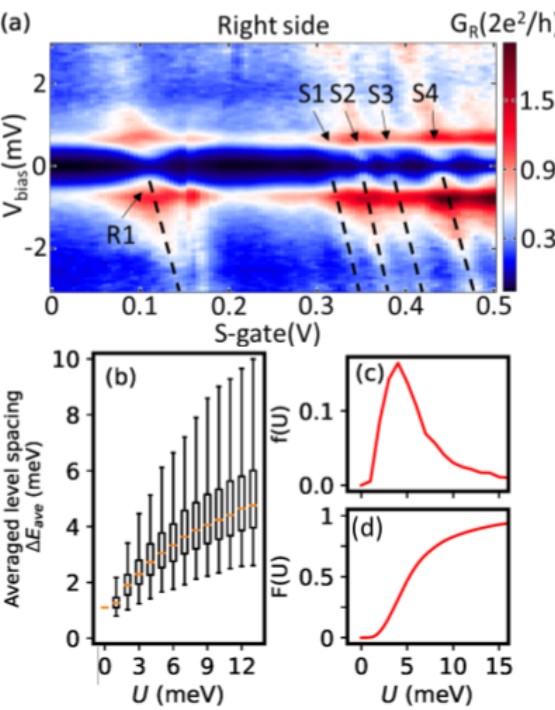

Figure 6: (a) Differential conductance $G_R$ as a function of bias voltage $V_{bias}$ and S-gate voltage at zero magnetic field. Fitting the resonances (black dashed lines) allows us to extract the level spacing between delocalized states S1-S4. (b) Statistics of the averaged level spacing $\Delta E_{ave}$ of the first four delocalized states in a nanowire of length $L = 400$ nm as a function of disorder strength $U$. Statistics for each value of $U$ is for 2000 disorder realizations. Orange line indicate the median, while the boxes correspond to the 25-75% range, and the whiskers indicate the 2nd and 98th percentiles. (c) Probability distribution $f(U)$ of disorder strength $U$ given the experimental averaged level spacing of $\Delta E_{ave} = 2.6$ meV. (d) Cumulative distribution $F(U) = \int_0^U f(U') \, dU'$, which indicates the chances of having a disorder strength less than $U$.

are localized by disorder, only partially separated quasi-Majorana modes can emerge. These generate uncorrelated near-zero energy features at the ends of the wire, and they cannot be unambiguously distinguished from trivial non-Majorana states. To realize genuine, well separated zero energy Majorana modes, disorder has to be reduced below a threshold. Ref. [35] classified the range of $U \gtrsim 1$ meV as the intermediate/strong disorder regime where the presence of well-separated MZMs at the edges of the system occurs in a very small faction of devices. Our results indicate that our current devices are deep within this regime, indicating that disorder needs to be significantly reduced in order to realize MZMs. Importantly, however, the method introduced here of estimating disorder strength from the delocalized state level spacing allows progress towards this goal to be tracked. After all, the method does not require MZMs to be realized to show that disorder has been reduced.

What are the consequences of having such a disorder strength for realizing Majorana zero modes in this device? To realize genuine, well-separated zero-energy Majorana modes, disorder has to be reduced below a certain threshold. Based on the analysis in Ref. [35], the estimated value of this threshold is $U \approx 1$ meV. For stronger disorder, the probability of realizing well-separated MZMs at the opposite edges of the system is very small. These results suggest that the device investigated in this work is deep within the strong disorder regime, hence the observation of correlated Majorana features is highly unlikely. Most importantly,

however, the method proposed here for estimating the disorder strength (based on the level spacing between delocalized states) can be an important tool for accessing key information regarding disorder, which is otherwise extremely difficult to acquire.

In conclusion, we demonstrate that the presence of delocalized and localized states in hybrid nanowires can be identified by measuring the dependence of left and right conductances on the gate voltages in a three-terminal geometry and establishing the presence (or absence) of end-to-end correlations. Simulations using a one-dimensional effective model suggest that the absence of low-energy correlated features at finite fields is the result of strong disorder-induced localization of states corresponding to the bottom of the topmost occupied subband. We also propose a method for estimating the disorder strength characterizing the device based on a statistical analysis of the level spacing between delocalized states. The estimated disorder strength is well above the limit consistent with the realization of Majorana zero modes. Future advances in materials and fabrication may help reduce the disorder strength, which is a prerequisite for enabling the realization of genuine, well-separated Majorana zero modes.

### Further Reading

Details about nanowire growth can be found in [36]. More information about Majorana zero modes and Andreev bound states in nanowires is discussed in [21,37,38]. More three-terminal geometry measurements in nanowires are reported in [29,39,40].

### Methods

Nanowire growth: Metalorganic vapour-phase epitaxy is used to grow the InSb nanowires used in this work. Devices are made from nanowire with 3-5 $\mu$m length and 120-150 nm diameter. Fabrication: InSb nanowires are manually transferred from the growth chip to the device chip, which has prefabricated bottom gates, using a micromanipulator. Contact patterns are written using electron beam lithography. In the first lithography cycle, superconducting contact (5 nm NbTi and 60 nm NbTiN) is sputtered onto the nanowire with an angle of 60 degree regarding the chip substrate. In the second lithography cycle, 10 nm Ti and 100 nm Pd is evaporated as normal contacts. Sulfur passivation followed by a gentle argon sputter cleaning is used to remove the native oxide on the nanowire before metal deposition.

Measurements are performed in a dilution refrigerator with multiple stages of filter at a base temperature of 40 mK. Standard low-frequency lock-in technique (77.77 Hz, 5 $\mu$V) is used to measure the devices. To remove the contribution from the measurement circuit, we normalized the differential conductance directly measured with the lock-in amplifiers, as described in Ref. [25].

### Volume and Duration of Study

To study the delocalized states and quantized ZBCP, 15 chips were fabricated and cooled down, on which more than 40 three-terminal devices were measured. Many of the devices had high contact resistance and were not studied in detail. About half of them were studied in detail, among which four devices showed delocalized states. For the device studied in this paper, more than 9000 datasets were obtained within three months.

### Data Availability

Data on several three-terminal devices going beyond what is presented within the paper is available on Zenodo (DOI 10.5281/zenodo.3958243).

## Author Contributions

G.B. and E.B. provided the nanowires. P.Y. and J.C. fabricated the devices. P.Y. performed the measurements. B.W. and T.S. performed numerical simulations. P.Y., B.W, T.S. and S.F. analyzed the results and wrote the manuscript with contributions from all of the authors.

## Acknowledgements

We thank S. Gazibegovic for assistance in growing nanowires.

**Funding information**    S.F. supported by NSF PIRE-1743717, NSF DMR-1906325, ONR and ARO. T.S. supported by NSF grant No. 2014156.

## Supplementary Information

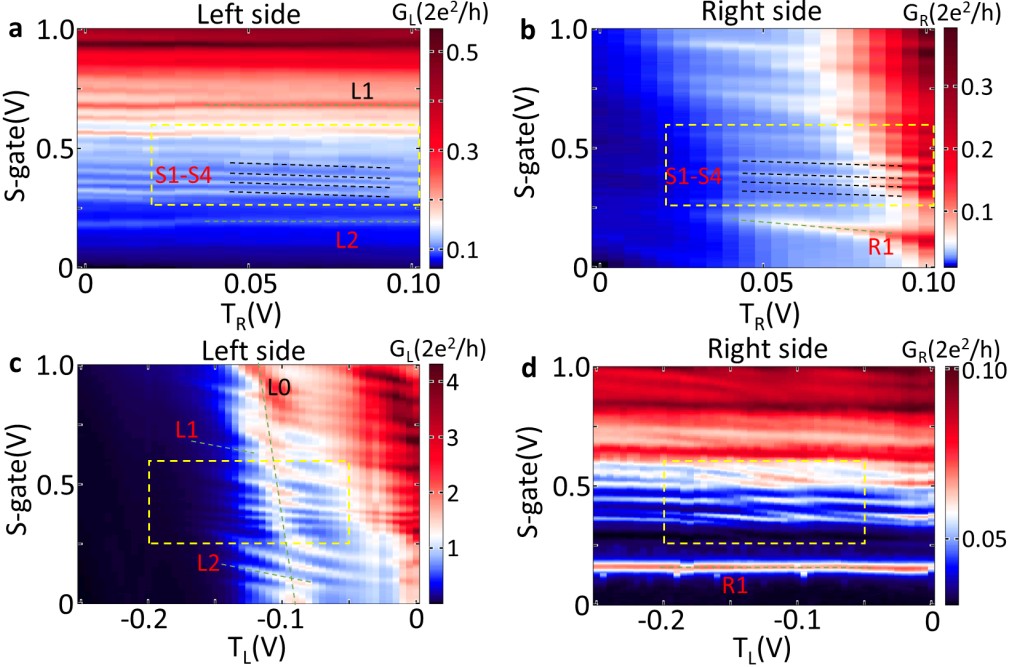

Figure S1: **Large scale barrier gates vs. S-gate scans.** (a) and (b) Zero-bias differential conductance $G_L$ and $G_R$ as functions of $T_R$ and S-gate voltage at zero magnetic field. Yellow rectangle indicates the region with delocalized states S1-S4. (c) and (d) Differential conductance $G_L$ and $G_R$ as functions of $T_L$ and S-gate voltage at zero magnetic field. L0, L1 and L2 are left localized states only appearing on the left side while R1 is right localized state visible on the right side. Note that the delocalized states only emerge for S-gate > 0.3 V.

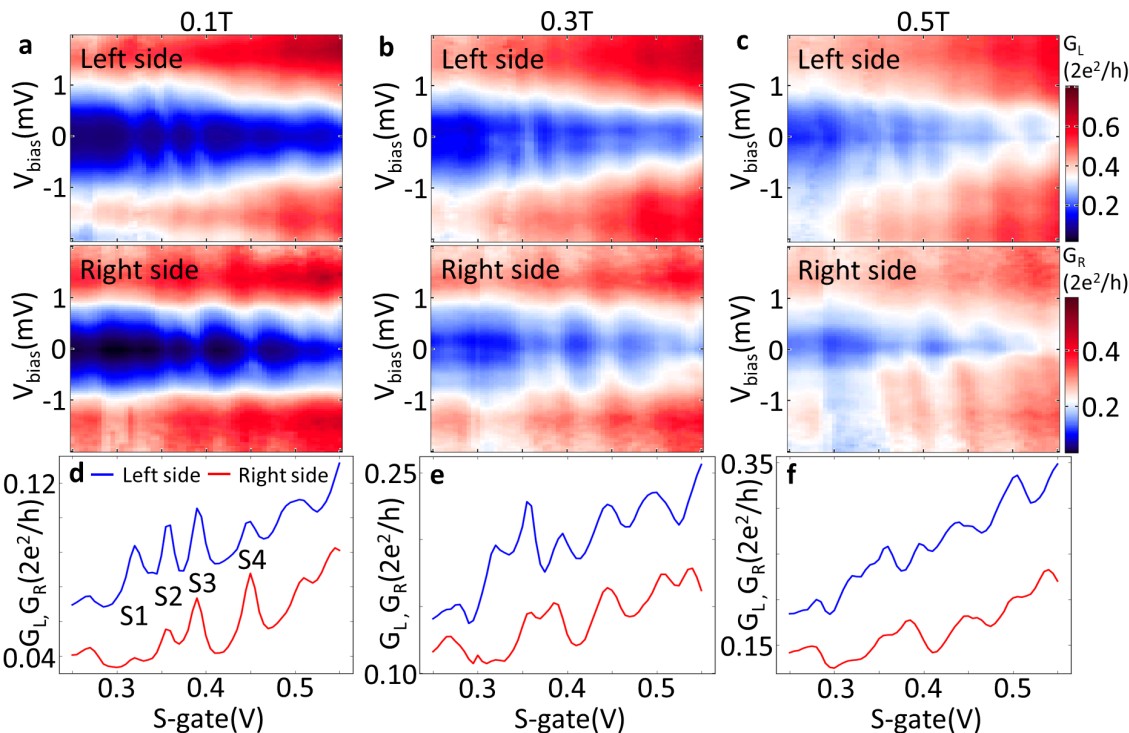

Figure S2: **Magnetic field dependence of the delocalized states.** (a),(b) and (c) Differential conductance $G_L$ and $G_R$ as functions of bias voltage and S-gate voltage at 0.1 T, 0.3 T and 0.5T. (d), (e) and (f) Zero-bias conductance traces from the two sides obtained from panels (a), (b) and (c). While delocalized states S1-S4 are traceable at 0.1 T between the two sides, the correlation is gradually lost at higher fields.

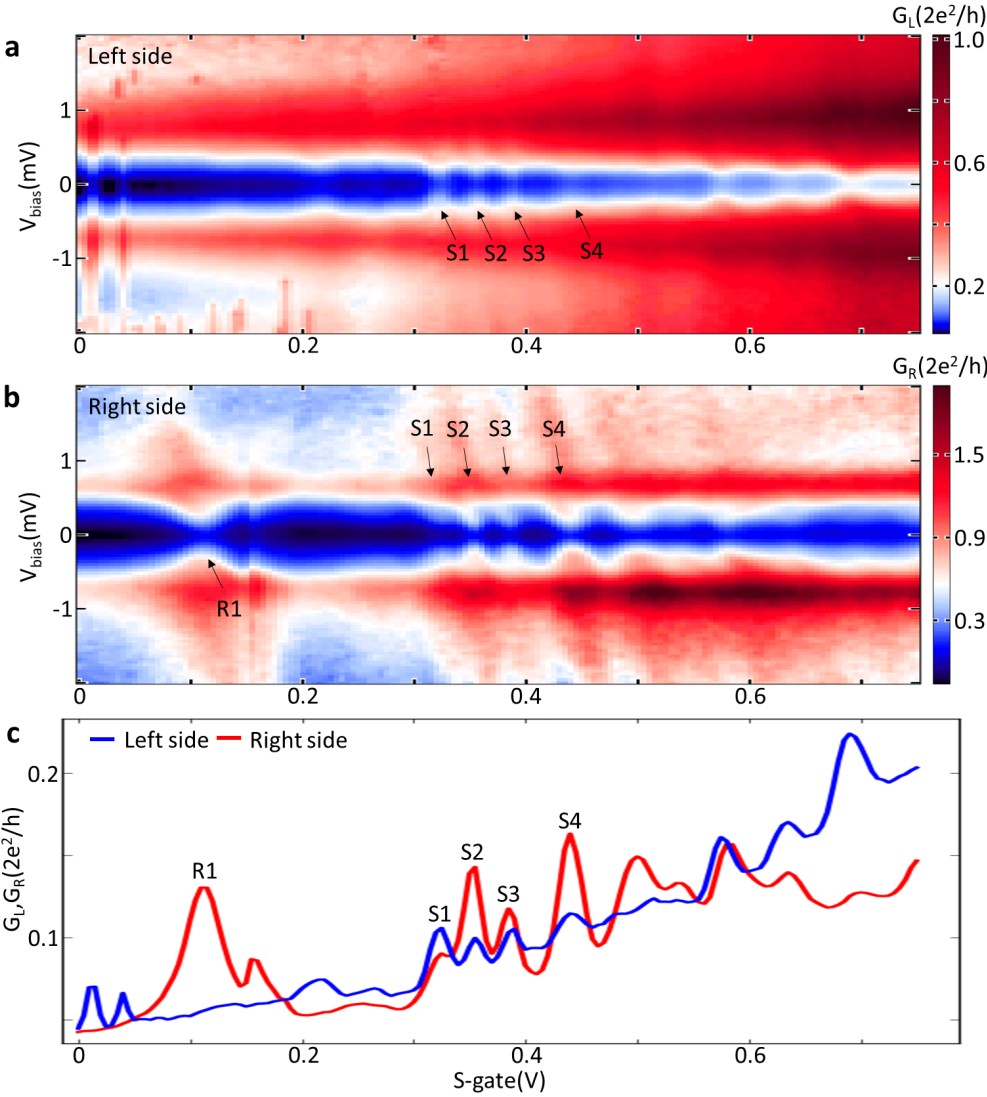

Figure S3: **Large range S-gate dependence of the delocalized states.** (a) and (b) Differential conductance $G_L$ and $G_R$ as functions of bias voltage and S-gate voltage at zero magnetic field. The delocalized states clearly extend above the induced gaps. (c), Zero-bias differential conductance $G_L$ and $G_R$ from (a) and (b).

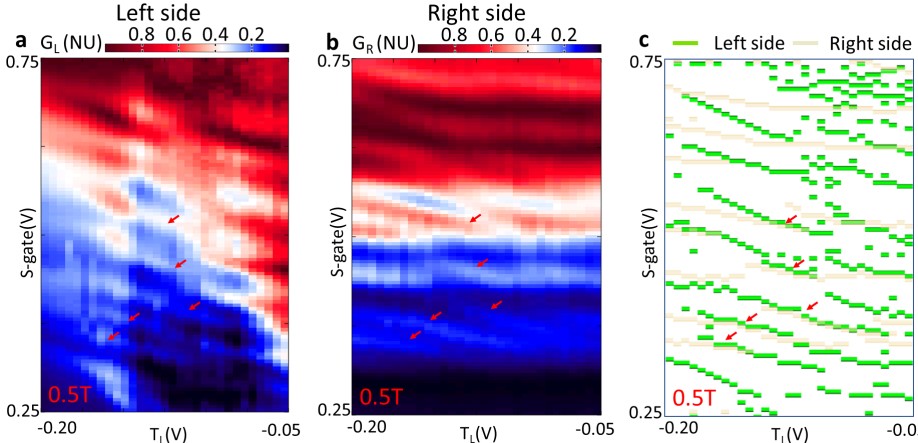

Figure S4: **Delocalized states at B = 0.5 T.** (a) and (b) Differential conductance $G_L$ and $G_R$ as functions of $T_L$ and S-gate voltage at B = 0.5 T. The correlation between the two sides is less apparent compared to zero field. However, delocalized states can still be traced in $T_L$ and S-gate scans. Correlated resonances are indicated by red arrows. Differential conductance in each row is in normalized units (NU) to improve the visibility. (c) Conductance maxima from panels (a) and (b).

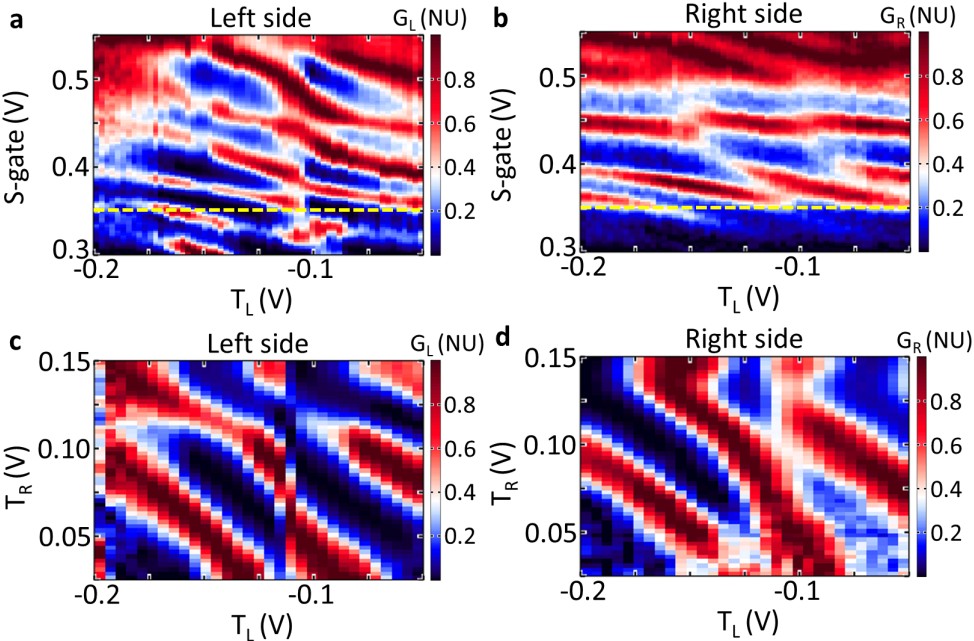

Figure S5: $T_L$ **vs.** $T_R$ **scans at zero magnetic field** (a) and (b) Zero-bias differential conductance $G_L$ and $G_R$ as functions of $T_R$ and S-gate voltage at zero magnetic field focusing on the regime with delocalized states. Yellow dashed lines indicate the S-gate setting for scans in Panels (c) and (d). (c) and (d) Zero-bias differential conductance $G_L$ and $G_R$ as functions of $T_R$ and $T_R$ at zero magnetic field. Delocalized states are not perfectly correlated due to irreproducible charge jumps affecting only the left side of the device thus likely occurring in the lead region. Differential conductance in each row is in normalized units (NU) to improve the visibility.

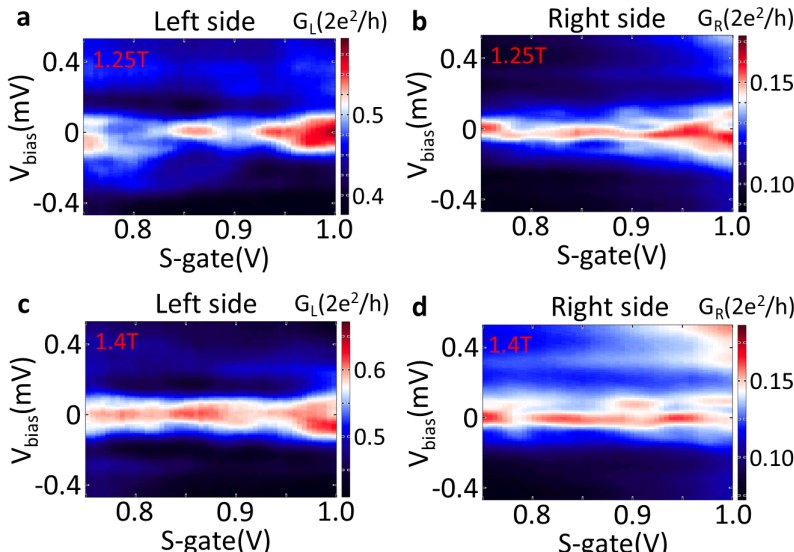

Figure S6: **Near-zero energy states at high magnetic fields** At high fields, near-zero energy states become ubiquitous on the left and right sides. As shown in panels (a), (b), (c) and (d), zero-bias peaks may accidentally appear on both sides at the same S-gate voltage. However, no correlation can be established after careful examination.

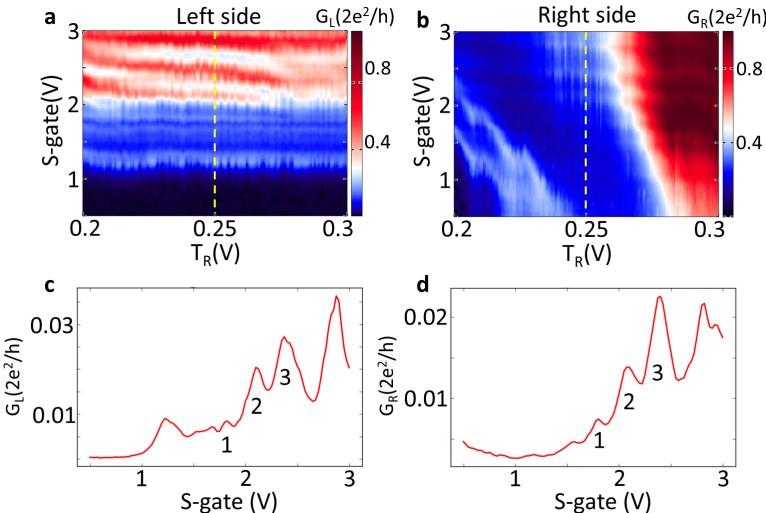

Figure S7: **Delocalized states in device B.** In another device with similar geometry, i.e. 400 nm nanowire proximitized segment, we also find delocalized states. As shown in (a) and (b), zero-bias differential conductance $G_L$ and $G_R$ as functions of $T_L$ and S-gate voltage show correlated resonances for S-gate > 1.5 V. (c) and (d), Linecuts taken from $T_L = 0.25$ V from panel (a) and (b) reveal three correlated conductance peaks, which are labeled as 1, 2, 3. Two states localized on the right side of the device is visible in the lower-left corner of panel (b). The magnetic field is zero for all the scans.

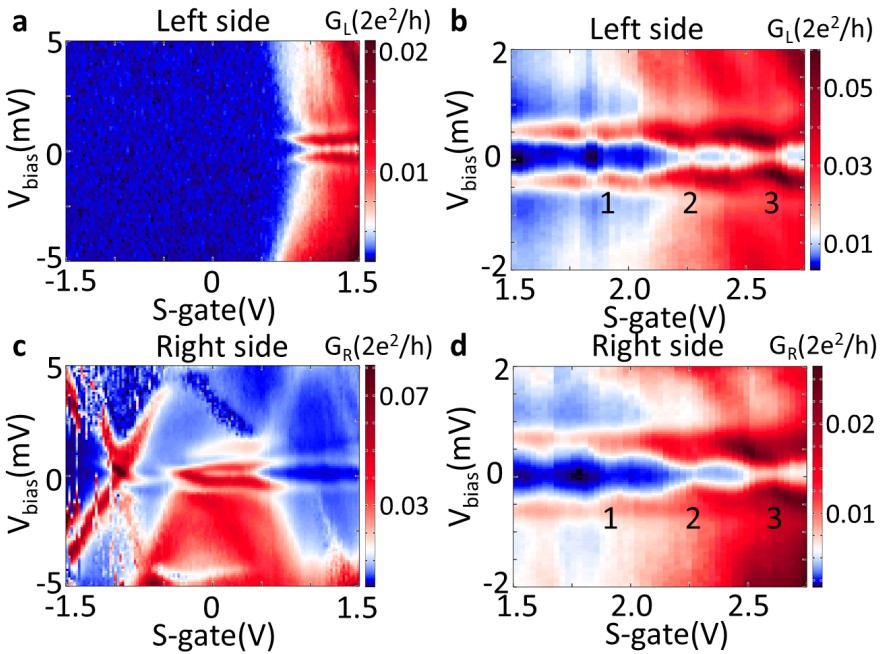

Figure S8: **Delocalized states and localized states in device B.** (a) and (c) Differential conductance $G_L$ and $G_R$ as functions of bias voltage and S-gate voltage. In this regime, -1.5V < S-gate < 1.5 V, the two sides show distinct features. (b) and (d) Differential conductance $G_L$ and $G_R$ in more positive S-gate regime. Three delocalized states, which also appear in Fig.S7, appear as correlated resonances on both sides. The fact that delocalized states appear beyond a certain positive S-gate voltage is consistent with the observations from device A.

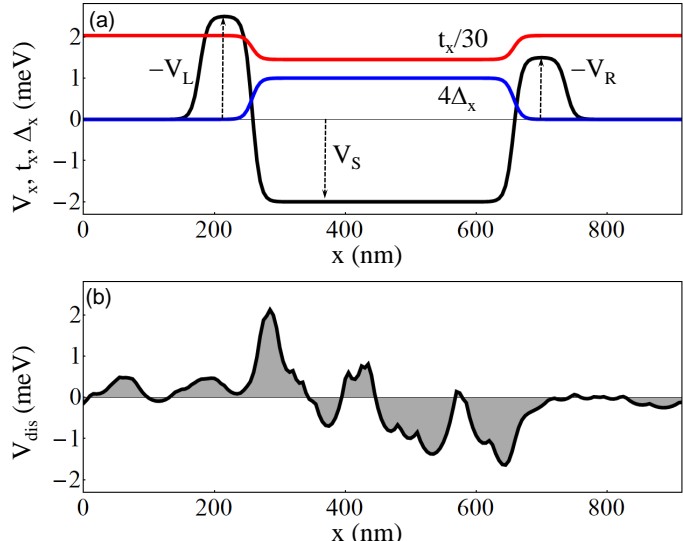

Figure S9: **Spatial dependence of the parameters used in the one-dimensional effective model.** (a) Position dependence of effective gate-induced potential (black line), nearest-neighbor hopping (red line), and induced superconducting gap (blue line). Note that the induced gap is $\Delta = 0.25$ meV in the superconducting region and vanishes in the normal regions. The hopping corresponds to an effective mass $m^* = 0.025 m_o$ in the normal regions and $m^* = 0.035$ in the superconducting region, with $m_o$ being the bare electron mass. (b) Effective disorder potential used in the numerical calculations.

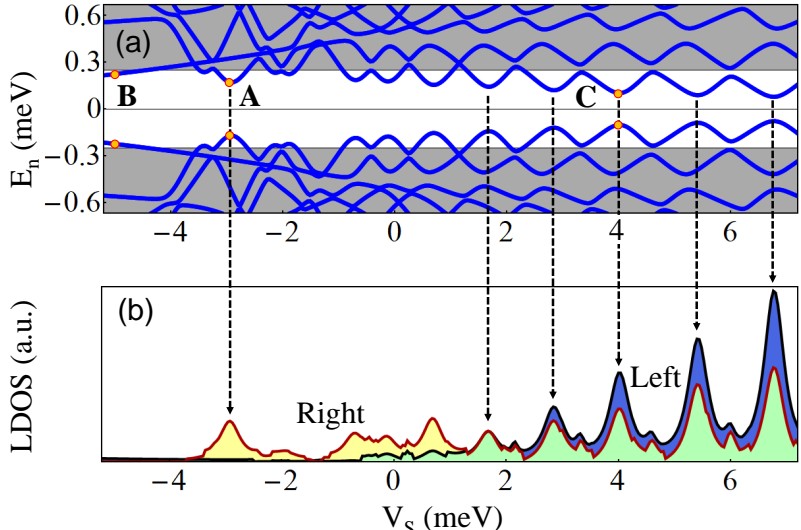

Figure S10: **Low-energy spectrum and zero-energy local density of states inside the normal regions as functions of the S-gate potential.** (a) Low-energy spectrum as a function of $V_s$ for fixed barrier potentials, $V_L = 0.25$ meV and $V_R = 0.75$ meV. (b) Zero-energy local density of states within the left (blue filled curve) and right (yellow filled curve) normal regions. Note that the "local density of states" is defined as $\sqrt{D_u D_v}$, where $D_u$ and $D_v$ are the particle and hole local densities of states, respectively. Also note that the local density of states was calculated assuming a finite spectral broadening $\eta = 30$ $\mu$eV. These results should be compared with the experimental findings shown in Fig. S3. The wave function profiles corresponding to the low-energy states labeled A, B, and C in (a) are shown in Fig. S11.

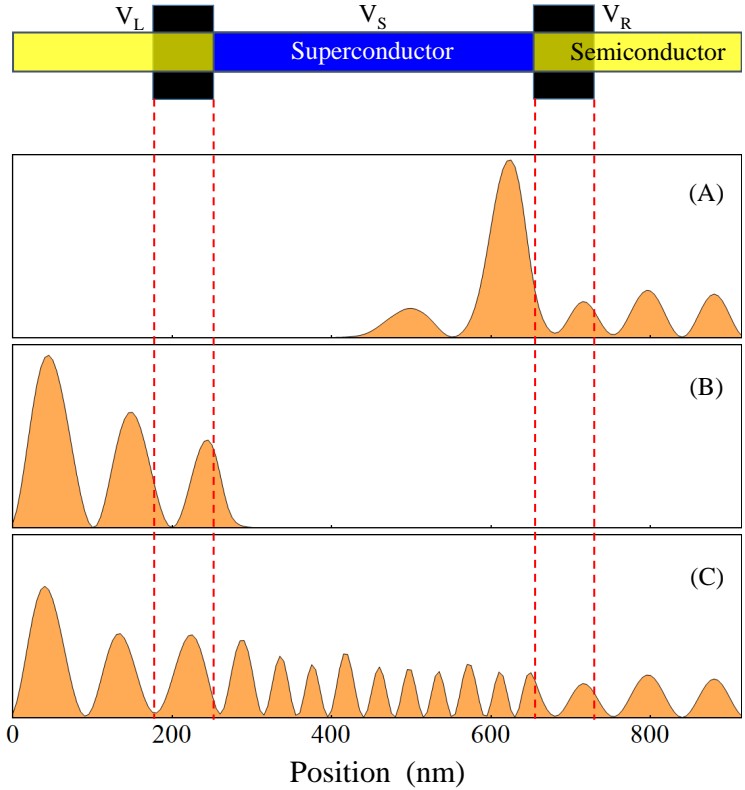

Figure S11: **Wave function profiles for different types of low-energy states.** Wave function probability densities corresponding to the states labeled A, B, and C in Fig. S10 (a). State A is localized near the right end of the superconducting region and penetrates into the adjacent normal region, which explains the peak in the (right) local density of states visible in Fig. S10 (b) at $V_s \approx -3$ meV. State B is localized almost entirely inside the left normal region. As a result, it has either particle or hole character and, consequently, generates a negligible "local density of states" $\sqrt{D_u D_v}$. State C is delocalized across the entire system due to its larger characteristic momentum, which is revealed by the highly oscillatory nature of this state. This type of delocalized states are responsible for the correlated peaks shown in Fig. S10 (b).

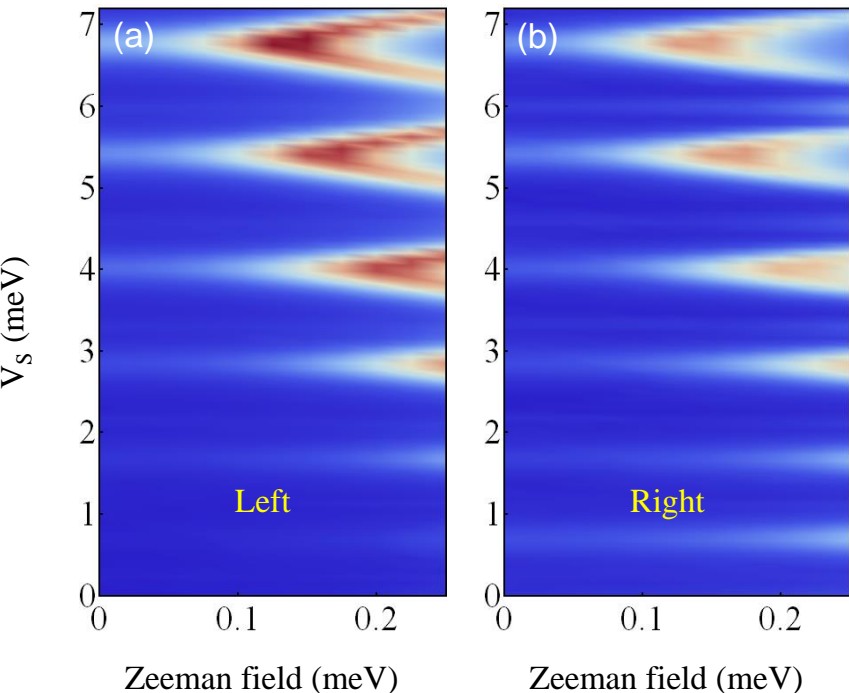

Figure S12: **Zero-energy local density of states as a function of Zeeman field and S-gate potential.** In the presence of a Zeeman field, the lowest-energy (in-gap) mode shown in Fig. S10 (a) spin-splits, with the lowest spin-split mode approaching and eventually crossing zero energy. This generates maxima of the zero-energy local density of states within $V_S$ windows that expand with the applied Zeeman field and eventually split. These results should be compared with the experimental findings shown in Fig. 3. Note that the vertical cuts at zero Zeeman field coincide with the curves shown in the lower panel of Fig. S10.

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
