# Peer review of "Delocalized states in three-terminal superconductor-semiconductor nanowire devices"

_SciPost Physics, doi:SciPost Phys. 15, 005 (2023)_

## Round 1 · Referee Report · Anonymous (Referee 1) · 2021-9-9

Report

In this joint experimental-theoretical work, the Authors discuss conductance measurements in a three-terminal InSb/NbTiN device. They identify bound states in the device via NS tunnelling conductance measurements and try to determine their spatial extent using non-local measurements between the normal terminals. These measurements, when compared to simulations of a toy model of the nanostructure, yield qualitative information about the inhomogeneity in the nanowire, which is deemed too high to allow for Majorana zero modes to occur in this material platform, at least in the current generation of devices.

This conclusion is sensible, and not surprising: NbTiN is sputtered on the InSb nanowire and devices thus fabricated consistently show a soft sub-gap density of states at B=0. This has been long identified as a non-starter for a gapped topological phase to occur. This conclusion is also not completely new: e.g. it has been previously reached by the same group in Phys. Rev. Lett. 123, 107703 (2019), based on NS measurements alone.

Repetita juvant, and in principle I am strongly in favour of another paper supporting this conclusion and providing further detail. However, I am not sure that the conclusion is well supported or illustrated by the data shown here. I have doubts related to the way the measurements were carried out, the interpretation of the data, the modelling, and the clarity of the presentation, which I think need to be sorted out before this work can be published. Below I present remarks on all these aspects separately.

REMARKS ON MEASUREMENT TECHNIQUE

The Authors conduct three-terminal lock-in measurements in the two configurations shown in Fig. 1a. In the configuration shown in black, a voltage bias is applied to the superconducting terminal and the resulting current response in the two normal terminals is measured to determine the conductances G_L and G_R. In the configuration shown in red, the superconducting terminal is floating, a voltage bias is applied to the right normal terminal, and the current response is measured on the left normal terminal to determine another conductance quantity named G_LR. The Authors associate G_L and G_R with the local density of states at the two NS junctions, and G_LR with the non-local transport between the two-terminals: according to this interpretation features (peaks) appearing in G_LR must be associated with states extended along the entire nanowire segment between the two NS junctions. However, I am not sure that this can actually be inferred from the data as analysed or presented.

First, in a three-terminal measurement, proper treatment of the resistor network and voltage divider corrections are essential in order to extract the right conductance matrix elements. This is described in detail in arXiv:2104.02671. Without this data processing step, local and non-local responses get mixed in the current measured when a lock-in excitation is applied to one of the three terminals. Given that the Authors do not give information on how the analysis of the three-terminal circuit is carried out, I worry that these spurious effects may affect their measurements carried out in the configuration shown in red in Fig. 1a.

Second, in the measurement configuration shown in black in Fig. 1a, the superconducting terminal is floating rather than grounded. This choice should be motivated: I believe it is not appropriate for the purpose of the experiment. In the absence of a ground between the two normal contacts, the applied bias will now be distributed across the two NS junctions. Therefore, the measurement will reflect a complicated convolution of two local density of states, rather than the proper off-diagonal element of a three-terminal conductance matrix. This is true in particular if both junctions are close to the tunnelling regime, which seems to be the case, rather than one closed and one open. As a consequence, I do not think that information about the extent of the bound states wave functions can be reached using this measurement.

These circumstances potentially make the discussion and interpretation of the data more complicated than portrayed in the manuscript. Thus, more details should be given on the circuit and the lock-in measurements: based on the information present, I cannot conclude that the measurements were carried out and processed correctly, where by “correctly” I mean in a way that allows one to infer spatial information from the measurements as done by the Authors. The Authors should provide more details on voltage divider effects and on the way the tunnel junctions are tuned in the second measurement configuration.

REMARKS ON THE INTERPRETATION OF THE DATA

In page 2, the Authors write:

"These states are common in our devices [14, 24]. We speculate that they may be located away from the semiconductor-superconductor interface and closer to the bottom of the nanowire. One caveat is that the bulk gap of NbTiN superconductor is much larger, of order 2 meV, so it is not possible to exclude Andreev reflections."

Given the supposed presence of extended states at zero energy and at zero magnetic field, the presence of semiconductor states not coupled to the superconductor make sense to me. However, per se this explanation does not require disorder: states at the nanowire bottom facets, away from the NbTiN, could be ballistic and still have zero energy. Later, in the theoretical analysis, these features are instead shown to exist in a disordered 1D model where the semiconductor states are all equally coupled to the superconductor. Thus, these two proposed scenarios seem to be inconsistent. What gives?

An alternative interpretation is not discussed: Is it possible that these conductance features are not a proxy of the density of states, but are instead weak Coulomb blockade oscillations caused by the spontaneous formation of quantum dots (density puddles) in the InSb nanowire?

In page 3, the Authors write:

"In this case the charging energy is reduced due to screening by the superconductor so that these are just wave functions with different spatial localisation within the nanowire."

This seems to be a speculative conclusion. If charging effects are present, even if with a reduced charging energy, then conductance features may be associated with Coulomb degeneracy points rather than energy levels of single-particle states.

Later in page 3, the Authors write:

"The presence of such states is expected given that the nanowire segment between the tunnel barriers is 400 nm in length, only a factor of 4 greater than the nanowire diameter. What is more surprising is that in the same segment we observe wave functions significantly localised either on the left or on the right end (L and R states)."

Can the localised wave functions actually be located not in the segment coupled to the superconductor, but only under the tunnel barriers, similar to the lead resonances identified in the theoretical modelling?

What is the expected induced coherence length of sub-gap states in this nanostructure? If the induced coherence length is not much shorter than 400 nm, there is no way to distinguish “localized” and “delocalized” states.

REMARKS ON THE MODELLING

  1. Can the Authors explain how they estimate Emin~ 5 meV from Fig. 5?

  2. Can the value of Emin be estimated from experimental data? This would, of course, be very valuable.

  3. I am puzzled by the presence of extended states (e.g. the one labeled by C in Fig. 5) at zero energy and zero magnetic field in this particular 1D model. Even in the presence of strong disorder, at B=0 the density of states at E=0 under the superconducting segment should vanish due to the s-wave pairing term, if the wire is long compared to the coherence length. Can the Authors explain? What is the induced coherence length in the simulation?

In general, despite the modelling effort and the extensive size of the study (9000 datasets as stated at the end of the paper), the lack of extraction of quantitative information from the experimental data is frustrating and affects the impact of this work.

REMARKS ON THE CLARITY OF THE PRESENTATION

  1. I am confused by the use of the term “delocalized state” in this work. It seems to be used to mean both “extended” (i.e., a state whose wave function has support throughout the entire nanowire segment, such as a proximitized semiconductor state with energy above the induced gap) and “non-local” (i.e., a Majorana state with wave function support only at the two ends of the nanowire, but not in the middle of the nanowire segment). This distinction is important, because to identify well-separated MZMs one must differentiate between extended and non-local wave functions. I wish the Authors adjusted the language used in the work to make these things more clear.

  2. Related to this point, the Authors write in page 1:

“Since Majorana modes are themselves delocalized states and require a uniform density along the nanowire, we focus on the regime with delocalized states to search for MZMs.“

It is not clear that one should focus on the regime with “delocalized” states to search for MZMs, as extended states may belong to different subbands and thus may not be a good proxy to find the right density range corresponding to a topological phase. A case in point, delocalised states are later associated with states located on the bottom facets of InSb, away from the superconductor. It seems to me that the logic is internally inconsistent on this.

  1. I wish the Authors did not use normalised units in Fig. 1, or at least, that they stated clearly what the normalisation is. This must be corrected.

I also wish that they did not use a strongly saturated and divergent colormap to present the data. These choices distort quantitative comparisons between data points and, in my opinion, go against best practices.

  1. In the introduction the Authors write:

“More generally, the three-terminal technique is a powerful method of studying the localisation of any wave functions, which is what we do in this work.”

Here, I believe that they should provide credit to the works that analysed three-terminal transport in hybrid nanostructures and elucidated this fact, such as (the list may not be exhaustive): Phys. Rev. B 97, 045421, Phys. Rev. B 103, 014513 and arXiv:2103.12217.

To conclude, the measurements and simulation shown in this work do not, in my opinion, provide useful evidence in favour of the conclusions reached. This paper focuses on specific features of the data which may not, by themselves, be related to the level of disorder in the device. Despite the abundance of data and simulations, no quantitative information regarding disorder strength (or its origin) is extracted. At the moment, referring to the criteria for publication in SciPost (https://scipost.org/SciPostPhys/about#criteria), I do find that this work currently does not meet any of the Expectations contained there, and that it fails to meet General Acceptance Criteria #1, 3, 4, 5. I hope that the Authors can leverage these remarks to improve their manuscript.

Requested changes

See report.

  • validity: low
  • significance: ok
  • originality: poor
  • clarity: ok
  • formatting: excellent
  • grammar: excellent

Author:  Sergey Frolov  on 2023-01-31  [id 3286]

(in reply to Report 2 on 2021-09-09)

This conclusion is sensible, and not surprising: NbTiN is sputtered on the InSb nanowire and devices thus fabricated consistently show a soft sub-gap density of states at B=0. This has been long identified as a non-starter for a gapped topological phase to occur.

A hard gap (at zero magnetic field) is an indicator of strong semiconductor-superconductor (SM-SC) coupling (for all states/bands), not an indicator of a disorder-free system. In other words, a device that exhibits a hard gap at zero magnetic field is perfectly consistent with the presence of strong disorder (e.g., in the semiconductor wire). A soft gap, is an explicit indication that disorder (at the interface or within the superconductor) is present, but not necessarily a worse starter for realizing Majorana physics. Note that in the strong semiconductor-superconductor coupling limit (which automatically leads to a hard gap) the critical (minimum) Zeeman field associated with the topological phase transition is given by the effective SM-SC coupling, which can be much larger than the induced gap. Consequently, in strongly coupled system (which are necessarily characterized by a hard hap), it may be difficult to reach the topological phase even in the absence of disorder/inhomogeneity. Also note that our device is not in the strong coupling limit (because the induced gap is significantly lower than the gap of the parent superconductor). Bottom-line, identifying starters/nonstarters based on “hardness” of the gap at zero magnetic field is just another misconception among many affecting this field.

This conclusion is also not completely new: e.g. it has been previously reached by the same group in Phys. Rev. Lett. 123, 107703 (2019), based on NS measurements alone.

This comment is not really helpful, because hardly any paper is completely new. However, in comparison with the 2019 PRL, we have improved device fabrication between that paper and this paper. And this paper includes a deeper analysis based on new simulations and proposes a new method for estimating the disorder strength, which may be useful in future studies.

REMARKS ON MEASUREMENT TECHNIQUE First, in a three-terminal measurement, proper treatment of the resistor network and voltage divider corrections are essential in order to extract the right conductance matrix elements. This is described in detail in arXiv:2104.02671.

It has not been the goal of our measurement to extract the conductance matrix. Instead, we infer the local vs nonlocal character of resonances from how they respond to gates. If the resonances shift with a gate that means the wavefunction has weight above that gate.

It may be interesting, in principle, to study in the future the more nuanced aspects of the nonlocal signals. Especially in the light of the overinterpretation of nonlocal signals like those in the preprint that the referee pointed to. But that would be the subject of another paper, while this paper is not about that.

Without this data processing step, local and non-local responses get mixed in the current measured when a lock-in excitation is applied to one of the three terminals. Given that the Authors do not give information on how the analysis of the three-terminal circuit is carried out, I worry that these spurious effects may affect their measurements carried out in the configuration shown in red in Fig. 1a.

We summarized our three-terminal measurement methods in the supplementary of https://www.nature.com/articles/s41567-020-01107-w The observations we present are robust, consistent with each other, as can be verified by checking also the full data on Zenodo. Again, we are essentially presenting two sets of local measurements. There is little ambiguity about the relation between the observed features and low energy states satisfying certain properties (essentially regarding the coupling to one or both leads).

Second, in the measurement configuration shown in black in Fig. 1a, the superconducting terminal is floating rather than grounded. This choice should be motivated: I believe it is not appropriate for the purpose of the experiment. In the absence of a ground between the two normal contacts, the applied bias will now be distributed across the two NS junctions. Therefore, the measurement will reflect a complicated convolution of two local density of states, rather than the proper off-diagonal element of a three-terminal conductance matrix.

The convolution is not complicated - as we see in Figure 1b, all states from the left and right sides show up when we measure from left-to-right in the two-terminal geometry. Again, there is no one type of nonlocal or another type of multiterminal measurement dictated ‘from above’. Our setup is fully described which allows for the independent interpretation of the data.

In the experience of one of us, SF, no matter how you wire up the multi-terminal device, you essentially get the same qualitative information. We have limited data to support that, available on Zenodo. We did not perform measurements in every possible configuration of current and voltage terminals for each of the studied regimes.

This is true in particular if both junctions are close to the tunnelling regime, which seems to be the case, rather than one closed and one open. As a consequence, I do not think that information about the extent of the bound states wave functions can be reached using this measurement.

We do not try to reach information about the extent of the states from the two-terminal measurement. This is obtained from the tunability by gates placed at different locations.

These circumstances potentially make the discussion and interpretation of the data more complicated than portrayed in the manuscript. Thus, more details should be given on the circuit and the lock-in measurements: based on the information present, I cannot conclude that the measurements were carried out and processed correctly, where by “correctly” I mean in a way that allows one to infer spatial information from the measurements as done by the Authors. The Authors should provide more details on voltage divider effects and on the way the tunnel junctions are tuned in the second measurement configuration.

The referee accurately describes our technique but with one omission. The referee does not say anything about the gate voltages. Yet, our technique is entirely based on comparing two things 1) which way the current flows and 2) which gates move the resonances. It is very simple and robust. Because we see states where the current flows to the left and the right gate does not move the resonances - that is a state located on the left. And so on.

There is a lot of great literature on nonlocal measurements and various schemes can be quoted as ‘correct’ or ‘best’. Some of us have done such measurements in the past. The fact is, our method and its data can be understood within its own description. The referee does not point to any specific flaw in the method, only to characterize it as ‘not correct’ and say that the full interpretation of all details of the signal can be more complicated. This is always the case. We focus on the most striking features - the gate dependence and its correlation with the path of current. And the referee does not point out any factors that are significant enough to invalidate our conclusions about where the states are localized.

To recap, (i) the features characterizing G_L (G_R) are necessarily connected with low-energy states that couple significantly to the left (right) lead and (ii) the spatial extent of these states is tested by varying different gate potentials along the wire. If a feature is sensitive (insensitive) to a certain potential, one can naturally conclude that the corresponding wave function extends (does not extend) inside the region affected by the potential. The procedure is straightforward and robust.

REMARKS ON THE INTERPRETATION OF THE DATA Given the supposed presence of extended states at zero energy and at zero magnetic field, the presence of semiconductor states not coupled to the superconductor make sense to me. However, per se this explanation does not require disorder: states at the nanowire bottom facets, away from the NbTiN, could be ballistic and still have zero energy. Later, in the theoretical analysis, these features are instead shown to exist in a disordered 1D model where the semiconductor states are all equally coupled to the superconductor. Thus, these two proposed scenarios seem to be inconsistent. What gives?

We have removed the speculative discussion of states not coupled to the superconductor. Because upon further looking at the data vs. the simulation we realized that the states that do hybridize with the superconductor can appear very close to the measured states.

An alternative interpretation is not discussed: Is it possible that these conductance features are not a proxy of the density of states, but are instead weak Coulomb blockade oscillations caused by the spontaneous formation of quantum dots (density puddles) in the InSb nanowire?

We do see clear Coulomb diamonds in the second device that we studied. See Figure S8. Compared to that, the resonances in the main text are rather blurred. They come for quantum dots with much more open barriers which quenches the charging energy. This is reasonable also given their proximity to NbTiN which is a huge chunk of metal. The quantum dot would have to be inside the S region (because the features are highly sensitive to the S gate, but not to the T_L/R voltages).

In page 3, the Authors write: "In this case the charging energy is reduced due to screening by the superconductor so that these are just wave functions with different spatial localisation within the nanowire." This seems to be a speculative conclusion. If charging effects are present, even if with a reduced charging energy, then conductance features may be associated with Coulomb degeneracy points rather than energy levels of single-particle states.

There can be a very small Coulomb energy that remains unquenched. But it is not worth exploring that given the other key energy scales are much larger. Broadening of lines means the barriers of a ‘dot’ are open and that dynamically quenches Coulomb energy.

Later in page 3, the Authors write: "The presence of such states is expected given that the nanowire segment between the tunnel barriers is 400 nm in length, only a factor of 4 greater than the nanowire diameter. What is more surprising is that in the same segment we observe wave functions significantly localised either on the left or on the right end (L and R states)." Can the localised wave functions actually be located not in the segment coupled to the superconductor, but only under the tunnel barriers, similar to the lead resonances identified in the theoretical modelling?

We think it is less likely in the experiment based on the gate settings. For example, state R1 disperses in gate TR much weaker than other fainter and unlabeled resonances that are essentially vertical and show no dependence on the S-gate. Those vertical resonances are due to the barrier for sure. And they are in contrast with the ones sensitive to S-gate.

What is the expected induced coherence length of sub-gap states in this nanostructure? If the induced coherence length is not much shorter than 400 nm, there is no way to distinguish “localized” and “delocalized” states.

This is a bit of a circular question. If we define coherence length as the length over which we find 67% of the wavefunction in one giant lump, then this experiment measures it to be clearly less than 400 nm (for the localized states).

There is no ‘expected’ coherence length. The characteristic length of sub-gap states depends on the strength of disorder/inhomogeneity and on the control parameters (e.g., the chemical potential/ S-gate potential).

REMARKS ON THE MODELLING 1. Can the Authors explain how they estimate Emin~ 5 meV from Fig. 5?

We have significantly re-worked the part of the paper related to disorder and redefined how we treat disorder in the model. We added Figure 6 and discussion with additional analysis. As a result, Emin no longer appears.

  1. Can the value of Emin be estimated from experimental data? This would, of course, be very valuable.

We agree with the referee that this would be very valuable. Indeed, this has motivated us to perform a new analysis that estimates the disorder strength present in the device. This new analysis is included as Fig. 6 and additional discussion.

  1. I am puzzled by the presence of extended states (e.g. the one labeled by C in Fig. 5) at zero energy and zero magnetic field in this particular 1D model. Even in the presence of strong disorder, at B=0 the density of states at E=0 under the superconducting segment should vanish due to the s-wave pairing term, if the wire is long compared to the coherence length. Can the Authors explain? What is the induced coherence length in the simulation?

The referee is correct that no states can exist at zero energy for B = 0 due to the s-wave pairing term. However, the state labeled C in Fig. 5 is NOT a zero energy state. They do not generate a zero-bias conductance peak (as function of the bias voltage). The reason why these states generate a signature in the zero-bias conductance is that the spectral features have finite broadening. In the simulation, we incorporate the effects of dissipation and finite temperature by including a finite spectral broadening, η= 30 μeV, which causes a small but finite density of states at zero-energy. For clarity, we have added the following two sentences on page 3:

We assume a finite spectral broadening, \eta = 30 micro eV. Note that this yields a non-zero density of states at zero energy, even though all states are gaped as a result of proximity-induced superconductivity.

Further clarification is provided by Figs. S10 (which explicitly shows the B=0 spectrum) and S12 (which illustrates the effect of broadening).

In general, despite the modelling effort and the extensive size of the study (9000 datasets as stated at the end of the paper), the lack of extraction of quantitative information from the experimental data is frustrating and affects the impact of this work.

We have now added Figure 6 and the related analysis in an attempt to extract quantitative information regarding the strength of disorder.

REMARKS ON THE CLARITY OF THE PRESENTATION 1. I am confused by the use of the term “delocalized state” in this work. It seems to be used to mean both “extended” (i.e., a state whose wave function has support throughout the entire nanowire segment, such as a proximitized semiconductor state with energy above the induced gap) and “non-local” (i.e., a Majorana state with wave function support only at the two ends of the nanowire, but not in the middle of the nanowire segment). This distinction is important, because to identify well-separated MZMs one must differentiate between extended and non-local wave functions. I wish the Authors adjusted the language used in the work to make these things more clear.

The distinction is important for theory. But in experiment we cannot tell the difference. What the referee calls extended would appear the same as nonlocal at the detection point. Only detailed investigation of the gate voltage dependences and other Majorana signatures will be able to provide some insight into this difference. In the revised manuscript we have included a sentence saying that the “delocalized” states “… are not non-local pairs of (localized) Majorana bound states or degenerate Andreev bound states.”

  1. Related to this point, the Authors write in page 1: “Since Majorana modes are themselves delocalized states and require a uniform density along the nanowire, we focus on the regime with delocalized states to search for MZMs.“ It is not clear that one should focus on the regime with “delocalized” states to search for MZMs, as extended states may belong to different subbands and thus may not be a good proxy to find the right density range corresponding to a topological phase. A case in point, delocalised states are later associated with states located on the bottom facets of InSb, away from the superconductor. It seems to me that the logic is internally inconsistent on this.

We removed the hypothesis that Majorana should be looked for in the regime with delocalized states. We explained in comments to referee 4 what was behind that idea. As already discussed, we removed the speculation that the measured states are decoupled from the superconductor.

  1. I wish the Authors did not use normalised units in Fig. 1, or at least, that they stated clearly what the normalisation is. This must be corrected.

In SG vs tunneling gate scan, the conductance changes a lot due to the tuning of tunneling gate. It is helpful and necessary to normalized the conductance in each row to make resonance clearer. We stated in the caption of Fig.2 about the normalization is for each row. We find the chosen way of plotting helpful, the referee does not explain why it is problematic. Any reader can download our data and plot them the way they prefer.

I also wish that they did not use a strongly saturated and divergent colormap to present the data. These choices distort quantitative comparisons between data points and, in my opinion, go against best practices.

For scans with huge conductance difference, we unfortunately have to sacrifice the visibility of unimportant features. To solve this problem, we uploaded all the raw data on Zenodo(DOI 10.5281/zenodo.3958243). Any reader can download our data and plot them the way they prefer.

  1. In the introduction the Authors write: “More generally, the three-terminal technique is a powerful method of studying the localisation of any wave functions, which is what we do in this work.” Here, I believe that they should provide credit to the works that analysed three-terminal transport in hybrid nanostructures and elucidated this fact, such as (the list may not be exhaustive): Phys. Rev. B 97, 045421, Phys. Rev. B 103, 014513 and arXiv:2103.12217.

References are not for giving credit, they are to help readers. We now add those references.

To conclude, the measurements and simulation shown in this work do not, in my opinion, provide useful evidence in favour of the conclusions reached.

We do not understand how this statement is justified after reading the comments. Again, the fundamental objective of this study is to correctly ascribe various observed features to specific types of microscopic states. None of the referee’s comments represents a serious challenge to our interpretation and none can provide a viable alternative scenario, as explained in detail above. We believe that the referee’s conclusion is unsubstantiated.

This paper focuses on specific features of the data which may not, by themselves, be related to the level of disorder in the device. Despite the abundance of data and simulations, no quantitative information regarding disorder strength (or its origin) is extracted.

We have added Figure 6 and surrounding discussion which attempts to relate the simulated level of disorder to the experimental features.

At the moment, referring to the criteria for publication in SciPost (https://scipost.org/SciPostPhys/about#criteria), I do find that this work currently does not meet any of the Expectations contained there, and that it fails to meet General Acceptance Criteria #1, 3, 4, 5. I hope that the Authors can leverage these remarks to improve their manuscript.

The referee does not point out in what way the manuscript does not meet the listed general acceptance criteria. We cannot act on what is implied. We find that it meets all of the criteria. The constructive aspects of the referee’s criticism have been incorporated in the revised manuscript, which, we hope, clarifies the main issues and substantially enhances the readability of the paper.

---

## Round 1 · Referee Report · Anonymous (Referee 2) · 2021-9-25

Report

In this experimental-theoretical work, the authors report three-terminal conductance measurements in a semiconductor-superconductor nanowire device. They observe quantum states that they appear as peaks in conductance and they aim at distinguishing between localized and delocalized states. They ascribe those states with the presence of disorder in the nanowire and develop a theoretical model to test the plausibility of their hypothesis.
This manuscript is a follow-up of their previous work [1]. They have measured the same device where they already distinguish localized and delocalized states in Figure 4 of [1]. In the present work, more data has been added in order to tackle the problem and another type of measurement has been performed, i.e. fig.1b.

Delocalized states might play an important role for the formation of MZMs in hybrid nanowires. Therefore, being able to detect them and to understand their origin is definitely important. However, I am not sure this follow-up work brings new insights to deserve publication in SciPost. Before this, I have some comments and questions for the authors.

In the following, I present my comments and questions.

Experimental set-up:
1. Why did the authors use the gates T3 to make the nanowire segments “highly electrostatically n-doped”? Would it not be easier/better to position the normal leads (NL and NR) closer to the S-lead? Is there a risk to create unwanted quantum dots (QDs) in the T3 segment?

2. Do the authors know if QDs are formed in the tunnel barriers? If yes, would it be possible to remove them by shrinking the size of the tunnel barrier as it has been shown in [2,3]? It is a known fact that QDs in the tunnel barrier affect the result of tunneling spectroscopy experiments. Therefore, getting rid of them would make the work stronger. Could the authors comment on this point?

General comments and questions:
1. Fig.1b represents the novelty of this work, however I am not fully sure that it is needed to reach the conclusion of this work. Why R1 and L1 are also present in this measurement even though they are localized on the right and left parts, respectively?

Could the authors reach the same conclusion without the result of Fig.1b?

Why is the S-lead floating and not grounded for this measurement? This is in contrast with what has been previously studied in the literature of three-terminal device, as an example see ref. [4]. Could the authors motivate and comment their choice? Do the authors expect differences if the s-lead would have been grounded instead of floating? In addition, a protocol explaining how to correct data properly in a three-terminal measurement has been recently published [5].

2. The abstract reads “We identify some states as delocalized above-gap states”, however this is not commented in the rest of the manuscript. One can understand that the delocalized states are above-gap states only looking at Fig.S3, is this statement correct? If so, could the authors explicitly mention it in the text?

3. MZMs are defined as delocalized. “Delocalized” means that the wave function is spread throughout all the semiconductor-superconductor system, whereas MZMs are more often described as “non-local” because the wave function is localized at the ends of the hybridized nanowire but there is no wave function in the middle. Could the authors explain more precisely what they define as delocalized?

Furthermore, the authors state “Since Majorana modes are themselves delocalized states and require a uniform density along the nanowire, we focus on a regime with delocalized states to search for MZMs.” Could the authors explain why MZMs should appear in the regime where there are delocalized states? Can the presence of delocalized states be detrimental for the formation of MZMs?

4. The localized and delocalized states seem to arise because of the disorder in the hybrid nanowire. Can other mechanisms give similar results? What is the difference between QD states and disorder states? Can the authors provide measurements in which superconductivity is suppressed, with a high perpendicular magnetic field for instance? I believe that such measurement would help to understand if there are QDs in the system.

5. In this work, quantum states appear in the tunneling spectroscopy experiment also in absence of magnetic fields. However, this is not the case for other work in the literature, see refs. [2,3,6,7]. Could the authors comment on the difference between their result and the result of refs. [2,3,6,7]? Please, see Fig.2b of ref. [2], Fig.1c of ref. [3], Fig.3b of ref. [6] and Fig.1b of ref. [7]. Is it a correct statement that in refs. [2,3,6,7] the level of disorder is much smaller?

6. It would be helpful to add the theoretical plot of Fig.S12 in Fig.3 of the main text.

7. Could the authors suggest how one could reduce the disorder in hybrid nanowires?

Simulations:
1. How do the authors estimate the amplitude of the disorder potential Emin?
2. Does the tunnel barrier length play a role in the results of simulations? If yes, could the authors explain how?

Minor comments:
1. Refs. 4, 30 and 31 has been published in peer-reviewed journals, please cite them properly.

2. In some cases, the measurements are plotted in normalized units. Could the authors give an explanation about this choice? Could they add the plots not in normalized units in the Supplementary Information?

3. I ask the authors to rephrase the following sentence “We fabricate three-terminal nanowire devices which nominally fulfill requirements for Majorana zero modes: they are built around InSb nanowires that have significant intrinsic spin-orbit coupling, with superconductivity induced by a NbTiN superconductor.” Coulomb blockade NbTiN island experiment never showed 2e transport at zero field and NbTiN exhibits a soft gap. Therefore, I am not sure that InSb/NbTiN fulfills the requirements for realizing MZMs. Furthermore, there are still many open questions, like what is the effective spin-orbit coupling of InSb coupled to NbTiN. For these reasons, I kindly ask the authors to rephrase the sentence.

4. Colorbar is missing for figure 5a,b,c,d. Y-axis label is missing for figure 5e.

5. It would be helpful to know the values of all gate voltages, for instance what is the value of TL in Fig.1b,c and d?

In conclusion, I believe that being able to distinguish localized and delocalized states via three-terminal conductance measurement will be important for future experiments. In this work, experiments and simulations give a strong suggestion that this is feasible and that there are delocalized states coming from disorder. However, I am not yet sure that this work in the present form deserves a publication in SciPost because I am not yet convinced that it adds valuable information hidden in their previous work [1]. In particular, the manuscript in the present form does not meet the General Acceptance Criteria 1,3 and 4 of SciPost.

References:
[1] Nat. Phys. 17, 482-488 (2021).
[2] Nat. Nano. 10, 232-236 (2015).
[3] Science 373, 82-88 (2021).
[4] Phys. Rev. Lett. 124, 036802 (2020).
[5] arXiv:2104.02671v1 (2021).
[6] Nat. Comm. 12, 4914 (2021).
[7] Nat. Nano. 13, 192-197 (2018).

Requested changes

See report.

  • validity: ok
  • significance: good
  • originality: low
  • clarity: ok
  • formatting: good
  • grammar: good

Author:  Sergey Frolov  on 2023-01-31  [id 3285]

(in reply to Report 3 on 2021-09-25)

Delocalized states might play an important role for the formation of MZMs in hybrid nanowires. Therefore, being able to detect them and to understand their origin is definitely important. However, I am not sure this follow-up work brings new insights to deserve publication in SciPost.

The clear idea of this paper is that - yes there are delocalized states in these nanowires. But only at very high densities and over a fairly short segment. This is a very important realization for the entire line of research. It puts the entire body of previous work in a new light. And it also points to where we need to be to renew making loud physics claims. This is the reason why we decided to write a dedicated paper about delocalized states and teamed up with theorists to understand this important regime. We have also added new analysis with this resubmission, see Figure 6, that connects experimental data to modeled disorder. We hope this enhances the usefulness of this work.

We argue that the paper satisfies this SciPost acceptance criterion: “Present a breakthrough on a previously-identified and long-standing research stumbling block;”. Now we realize that there are no delocalized states at the densities where people search for Majorana, which has been a stumbling block for a long time where people present data on 2 micron long nanowires and treat them as uniform quantum wires.

Before this, I have some comments and questions for the authors. In the following, I present my comments and questions.Experimental set-up: 1. Why did the authors use the gates T3 to make the nanowire segments “highly electrostatically n-doped”? Would it not be easier/better to position the normal leads (NL and NR) closer to the S-lead? Is there a risk to create unwanted quantum dots (QDs) in the T3 segment?

The gates T3 are introduced to increase fabrication yield. And indeed, this creates additional states in those segments. But as shown in Extended Data Fig. 5 in our previous paper (https://www.nature.com/articles/s41567-020-01107-w), the resonances we observed in S-gate scans show no considerable gate dependence on T3, indicating the associated wavefunctions live far away from T3.

More generally, over the years we have tried many variations of the designs. The general findings do not change. For instance, ‘unwanted’ states can form under the leads no matter how narrow the contact spacing is. In some experiments, such as ‘Ballistic superconductivity’ and ‘Ballistic Majorana’, the contact spacing is so small that the effect of the gates is screened, as a result the extra states that are still present couple weakly to the gates and create the impression of ‘clean’ devices. Our method so far has been not to eliminate them fully but to understand them and clearly separate their influence from the effects of the states living under the superconductor.

  1. Do the authors know if QDs are formed in the tunnel barriers? If yes, would it be possible to remove them by shrinking the size of the tunnel barrier as it has been shown in [2,3]? It is a known fact that QDs in the tunnel barrier affect the result of tunneling spectroscopy experiments. Therefore, getting rid of them would make the work stronger. Could the authors comment on this point?

Such wavefunctions are found in devices of various geometries. It has not been possible so far to eliminate “quantum dots” or localized wavefunctions from devices in a systematic way, that is by making a change to something like the gate geometry. Occasionally, devices with fewer or less obvious localized states are found. It is also possible to select the data in such a way that localized wavefunctions are not presented in the paper, which applies to some of the published works, and is an inappropriate data selection method.

Note that such “quantum dot” states also occur (quite naturally) in the simulation. As long as we understand what type of feature is associated with them, they are quite benign.

General comments and questions: 1. Fig.1b represents the novelty of this work, however I am not fully sure that it is needed to reach the conclusion of this work.

We disagree that this panel represents the novelty of this work. Fig.1b is meant to strengthen our conclusion regarding the presence of extended/delocalized states that couple to both leads. However, the interpretation of nonlocal conductance is less straightforward than the interpretation of local conductance features.

Why R1 and L1 are also present in this measurement even though they are localized on the right and left parts, respectively?

In Figures 1b we measure from left to right so we go through all the states in between.

Could the authors reach the same conclusion without the result of Fig.1b?

Without Fig.1b, the general conclusion would still be the same. Fig.1b is a confirmation of our three-terminal measurements results.

Why is the S-lead floating and not grounded for this measurement?

The signal is much smaller if you ground it. It is the same pattern just less current because some of the current goes to the S lead.

This is in contrast with what has been previously studied in the literature of three-terminal device, as an example see ref. [4]. Could the authors motivate and comment their choice? Do the authors expect differences if the s-lead would have been grounded instead of floating? In addition, a protocol explaining how to correct data properly in a three-terminal measurement has been recently published [5].

There is no one authoritative source of knowledge for how to perform nonlocal measurements. But furthermore, we are using a different scheme which is presented in the figure. In Figure 1 panel b it is a two-terminal measurement.

  1. The abstract reads “We identify some states as delocalized above-gap states”, however this is not commented in the rest of the manuscript. One can understand that the delocalized states are above-gap states only looking at Fig.S3, is this statement correct? If so, could the authors explicitly mention it in the text?

We remove ‘above-gap’ from the abstract. The characterization of these states as ‘above’ gap referred to their presence above the apparent induced gap feature. At the same time, they are clearest within the bulk gap of NbTiN. The bias voltage dependence can now also be found in the new Figure 6 in the main text.

  1. MZMs are defined as delocalized. “Delocalized” means that the wave function is spread throughout all the semiconductor-superconductor system, whereas MZMs are more often described as “non-local” because the wave function is localized at the ends of the hybridized nanowire but there is no wave function in the middle. Could the authors explain more precisely what they define as delocalized?

Operationally, “delocalized” and “not local” are equivalent. Indeed, we cannot tell the difference between different types of non-local states based on measurements at the two ends of the system, as both types of states generate end-to-end features. Note, however, that the simulations provide additional information. For example, based on the qualitative agreement between experiment and simulation, we argue that the finite energy states that generate end-to-end correlated features at zero field are delocalized states extending throughout the whole device and coupling strongly to both leads. For clarity, we have included in the text a sentence saying that the “delocalized” states “… are not non-local pairs of (localized) Majorana bound states or degenerate Andreev bound states.”

Furthermore, the authors state “Since Majorana modes are themselves delocalized states and require a uniform density along the nanowire, we focus on a regime with delocalized states to search for MZMs.” Could the authors explain why MZMs should appear in the regime where there are delocalized states? Can the presence of delocalized states be detrimental for the formation of MZMs?

We removed this statement from the paper. See discussion with Referee 4. The presence of delocalized states is not detrimental to the emergence of Majorana physics (all bulk states are delocalized in the clean limit). However, their presence could indicate large localization length scales for all states, relative to the system syze (which is not the case in the device studied here). In this scenario, if MZMs are present, they could overlap strongly, which is detrimental to their topological protection. Simply using longer wires would solve this problem.

  1. The localized and delocalized states seem to arise because of the disorder in the hybrid nanowire. Can other mechanisms give similar results? What is the difference between QD states and disorder states? Can the authors provide measurements in which superconductivity is suppressed, with a high perpendicular magnetic field for instance? I believe that such measurement would help to understand if there are QDs in the system.

As we use high critical field superconductor NbTiN, it is impractical to fully suppress the superconductivity.

People often associate ‘quantum dots’ with objects that have charging energy, for historic reasons. In this paper we use the language of localized states to encompass any such states regardless of their origin and nature. They can have charging energy or not, they can derive from superconductivity or not. Referring to these states as ‘quantum dot states’ would also be fine, but it would generate a lot more confusion.

  1. In this work, quantum states appear in the tunneling spectroscopy experiment also in absence of magnetic fields. However, this is not the case for other work in the literature, see refs. [2,3,6,7]. Could the authors comment on the difference between their result and the result of refs. [2,3,6,7]? Please, see Fig.2b of ref. [2], Fig.1c of ref. [3], Fig.3b of ref. [6] and Fig.1b of ref. [7]. Is it a correct statement that in refs. [2,3,6,7] the level of disorder is much smaller?

We do not see fundamental differences between any of the published results and our work. If the entire bodies of data were to be compared, this would become clear. Note that the correlated features observed at zero magnetic field are not due to zero-energy states; they are generated by finite energy states in the presence of finite spectral broadening induced by the coupling to the metallic leads.

  1. It would be helpful to add the theoretical plot of Fig.S12 in Fig.3 of the main text.

We thank the referee for the suggestion, but we do not wish to combine theoretical and experimental plots in the same figures in this paper. We would like to keep it clear that we are not fitting the data with the model.

  1. Could the authors suggest how one could reduce the disorder in hybrid nanowires?

That is a billion-dollar question. We think improvements are possible. And any improvement, even incremental, will bring more clarity to the measurements we do here. Can we observe delocalized states in a 600 nm or a 1-micron long nanowire? At what energy (i.e., gate potential values)? In principle, if low-enough energy states become effectively delocalized at zero magnetic field, they will lead to the emergence of MZMs at some finite magnetic field.

In addition, we remark that our method introduced in the resubmitted version of this manuscript for estimating the disorder strength (see Figure 6 and surrounding text) should be useful going forward in monitoring the reduction of disorder in hybrid nanowires. Indeed, it seems promising in monitoring progress in disorder reduction (since there may be several steps before disorder levels consistent with the presence of Majorana zero modes are reached).

Simulations: 1. How do the authors estimate the amplitude of the disorder potential Emin?

We have significantly re-worked the part of the paper related to disorder and redefined how we treat disorder in the model. We added Figure 6 and discussion with additional analysis. As a result, Emin no longer appears in the discussion.

  1. Does the tunnel barrier length play a role in the results of simulations? If yes, could the authors explain how?

Yes, the tunnel barrier length plays a role in the simulation results since the length and barrier heights (VL and VR) control the coupling between the proximitized region and lead regions. This, in turn, determines the visibility of various features within the conductance data. The total length of the normal region also affects the results since states with significant weight in those regions have a smaller induced gap (and higher visibility at zero bias). We note, however, that while the exact value chosen for the tunnel barrier length quantitatively affects our results, the qualitative features will remain unchanged (for example the local density of states in Fig. 5 will still show delocalized and localized states and lead resonance for reasonable changes to the tunnel barrier length).

Minor comments: 1. Refs. 4, 30 and 31 has been published in peer-reviewed journals, please cite them properly.

We have made these requested changes.

  1. In some cases, the measurements are plotted in normalized units. Could the authors give an explanation about this choice? Could they add the plots not in normalized units in the Supplementary Information?

Using normalized units is to make the states more visible. Since in SG vs tunneling gate scans the conductance varies a lot due to the tuning of the tunneling gate, we have to normalize the conductance in each row to clearly show the resonances. Full original raw data are available with the paper and can be plotted in any other way that a reader prefers.

  1. I ask the authors to rephrase the following sentence “We fabricate three-terminal nanowire devices which nominally fulfill requirements for Majorana zero modes: they are built around InSb nanowires that have significant intrinsic spin-orbit coupling, with superconductivity induced by a NbTiN superconductor.”

There are uncertainties as to whether any of the systems studied to date fulfil all Majorana requirements. The word ‘nominally’ takes care of that. The “standard” requirements involve three key ingredients: superconductivity, spin-orbit coupling, and Zeeman splitting. The fabricated devices incorporate (at least nominally) these key ingredients. However, there is an additional critical requirement (typically not mentioned explicitly in the literature): the system has to be uniform within an energy scale less than (or, at most, comparable to) the “Majorana energy scale” E_M. This study explicitly demonstrates that the uniformity requirement is not satisfied by the device used in the experiment.

Coulomb blockade NbTiN island experiment never showed 2e transport at zero field and NbTiN exhibits a soft gap. Therefore, I am not sure that InSb/NbTiN fulfills the requirements for realizing MZMs.

2e Coulomb blockade or hard gap are not required to realize MZM. They are only required to study the coherent evolution of MZM. In particular, the absence of a soft gap (i.e., a hard gap) is essential to the topological protection of MZMs, but is not a requirement for their emergence.

Furthermore, there are still many open questions, like what is the effective spin-orbit coupling of InSb coupled to NbTiN. For these reasons, I kindly ask the authors to rephrase the sentence. It is about the same. Because the g factor is unchanged in the presence of deposited NbTiN. The bottom line revealed by this study is that, regardless of how well the three “standard conditions” are satisfied (there is definitely induced superconductivity and strong Zeeman effect in the device; very likely the spin-orbit coupling is significant), the uniformity requirement is definitely not satisfied. There is no evidence that this requirement is satisfied in other types of devices (e.g., epitaxial structures). Consequently, progress toward Majorana physics necessarily involves reducing nonuniformity/disorder in the system. In this context we note that numerical simulations show that weak spin-orbit coupling leads to longer localization length scales; this scenario is unlikely, considering the length of our device.

  1. Colorbar is missing for figure 5a,b,c,d. Y-axis label is missing for figure 5e.

We thank the referee for pointing out this omission. We’ve added a colorbar for Fig. 5 a,b,c, and d and added y-axis labels for Fig. 5. e

  1. It would be helpful to know the values of all gate voltages, for instance what is the value of TL in Fig.1b,c and d?

We agree the gate voltage values should be placed in the text. We have now added them to the main figures.

---

## Round 1 · Referee Report · Anonymous (Referee 3) · 2021-9-27

Report

The manuscript under review aims to be a systematic study of localized and delocalized states in short proximitized nanowires. To this end, the authors perform local and non-local conductance measurements to identify those states, and show a comparison to numerical simulations.

This paper is follow-up study to an earlier Nature Physics paper of the same experimental group, but focuses on a different parameter regime (different super-gate voltage) and different measurements (non-local conductance measurements), and hence is a sufficiently different study to be considered as a stand-alone research.

However, the current version of the manuscript feels rather uninspired and it is not clear to me what we can really learn from these measurements. In addition, it contains several claims that are in my opinion unfounded and not supported by the research. I will list specific points below.

In summary, I believe that the measurements in this paper definitely deserve publication somewhere, and after the controversial statements I list below have been amended, I think the paper could be published for example in SciPost Physics Core. Given that a clear breakthrough/new idea is missing in this paper (local and non-local measurements have been used before), I currently don't think that publication in SciPost Physics is feasible.

Specific questions/criticism:

  1. The authors write "In most cases, to identify non-Majorana states it is sufficient to analyze two-terminal measurements on just one nanowire in a self-consistent fashion." What do they mean by that? That one can identify trivial ABS always by two-terminal mesaurements? What do they mean by "self-consistent" in this context?

  2. On page 2, the authors speculate on the origin of the delocalized states at zero field. They argue that because the states are inside and outside the gap, it is unlikely that they arise from Andreev reflection, and thus need to be far away from the superconductor. The authors then later take this premise as granted. I am missing a more critical treatment, also discussing other possibilities. As the authors write themselves, this is a speculation from their side.

  3. On page 3, the authors write "What is more surprising is that in the same segment we observe wavefunctions significantly localized either on the left or on the right end (L and R states)." First, do the authors know what states those are? Is there a strong coupling to the superconductor, or are these mainly localized in a dot and weakly coupled to the superconductor? Second, why is the presence of these states surprising? Many experiments have seen quantum dot states near tunnel barriers.

  4. Further on the same page, the authors write "In particular, we explore the hypothesis that correlated subgap resonances can be found in the regime that shows delocalized wavefunctions at zero field.". I do not understand this statement, and I strongly believe it to be wrong. First, ideally one should look for Majorana zero modes (MZMs) in a regime where there is a good superconducting gap in the absence of a magnetic field, and hence there should be no delocalized states. Furthermore, the authors essentially speculated before that the delocalized states have little induced superconductivity - again detrimental for MZMs.

  5. On page 4, the authors argue that for localized states strong disorder is necessary. Why is this the case? Isn't this due to the limitations of the model, i.e. the choice of potentials, etc.? I can imagine to get quantum dot like states even in the absence of disorder if potentials are chosen accordingly.

  6. My biggest concern is the applicability of the numerical simulations to the experiment: The authors argued before that the delocalized states are states separated from the superconductor. However, in the 1D numerical model there is just one mode that is well-proximitized. That is, in my opinion, why the numerical simulations need strong disorder to get agreement with experiment. But this conclusion is weak, having started from orthogonal assumptions!

  7. I have a general question: How general are the experimental results for hybrid nanowire systems? In particular, the experiments are performed with sputtered NbTiN which is presumably very disordered. What would you expect to change with e.g. epitaxial superconductors?

  • validity: low
  • significance: ok
  • originality: ok
  • clarity: low
  • formatting: reasonable
  • grammar: good

Author:  Sergey Frolov  on 2023-01-31  [id 3284]

(in reply to Report 4 on 2021-09-27)
Category:
remark
answer to question
objection
reply to objection
validation or rederivation
pointer to related literature

However, the current version of the manuscript feels rather uninspired and it is not clear to me what we can really learn from these measurements.

In summary, I believe that the measurements in this paper definitely deserve publication somewhere, and after the controversial statements I list below have been amended, I think the paper could be published for example in SciPost Physics Core. Given that a clear breakthrough/new idea is missing in this paper (local and non-local measurements have been used before), I currently don't think that publication in SciPost Physics is feasible.

We thank the referee for ideas and suggestions. At the same time, we note that the “uninspired” comment is not very helpful. We hope, however, that the clarifications incorporated into the revised version of the manuscript and the newly proposed method for estimating the disorder strength in the device will change this feeling.

The key physical picture that emerges from this study is that the hybrid wire supports both localized and delocalized states, but all low-energy states that (could) generate Majorana-like zero-bias features are highly localized near one of the ends of the system and, consequently, do not generate end-to-end correlations. This is a very important realization for the entire line of research. It puts the entire body of previous work (based on this type of device) in a new light and suggest the presence of strong disorder/inhomogeneity.

In this resubmission, we clarified our analysis and, most importantly, used the measurements to estimate the effective disorder/inhomogeneity strength in the system. We have added Figure 6, which contains these new results.

We argue that the paper satisfies this SciPost acceptance criterion: “Present a breakthrough on a previously-identified and long-standing research stumbling block;”. Now we show explicitly that there are no delocalized states within the parameter regime expected to support Majorana zero modes, most likely as a result of strong disorder. The absence of (correlated) Majorana features has been a stumbling block for a long time, with people presenting data on 2 micron long nanowires and treating them as uniform quantum wires.

  1. The authors write "In most cases, to identify non-Majorana states it is sufficient to analyze two-terminal measurements on just one nanowire in a self-consistent fashion." What do they mean by that? That one can identify trivial ABS always by two-terminal mesaurements? What do they mean by "self-consistent" in this context?

We rephrase this idea: “In most cases, to identify non-Majorana states it is sufficient to analyze two-terminal measurements by testing the features associated with tunneling into one end of the nanowire against specific Majorana signatures, such as stability with respect to local gate potentials.”

  1. On page 2, the authors speculate on the origin of the delocalized states at zero field. They argue that because the states are inside and outside the gap, it is unlikely that they arise from Andreev reflection, and thus need to be far away from the superconductor. The authors then later take this premise as granted. I am missing a more critical treatment, also discussing other possibilities. As the authors write themselves, this is a speculation from their side.

We removed the speculative discussion in the resubmitted version. Upon studying the numerical results from our model, we conclude that states which do hybridize with the superconductor can generate features that look exactly like those measured experimentally (under the natural assumption that there is a finite level broadening, consistent with the experimental conditions).

  1. On page 3, the authors write "What is more surprising is that in the same segment we observe wavefunctions significantly localized either on the left or on the right end (L and R states)." First, do the authors know what states those are? Is there a strong coupling to the superconductor, or are these mainly localized in a dot and weakly coupled to the superconductor? Second, why is the presence of these states surprising? Many experiments have seen quantum dot states near tunnel barriers.

States L1 and R1 (Fig. 1) are strongly tunable with the S-gate. Their sensitivity to S-gate is similar to the sensitivity of tunnel states. They are labeled L and R because they appear in G_L and G_R, respectively. Through the combination of their sensitivity to S-gate and their manifestation in only one of the local conductances, we conclude that these states are localized under the superconductor, closer to one side of the S-segment. Their presence is surprising because the S-segment is rather short, 400 nm. On the other hand, the possibility of having such states is supported by the numerical calculations (A state in Fig 5).

  1. Further on the same page, the authors write "In particular, we explore the hypothesis that correlated subgap resonances can be found in the regime that shows delocalized wavefunctions at zero field.". I do not understand this statement, and I strongly believe it to be wrong. We removed this hypothesis, which was very natural from the experimental point of view, but does not appear to be helpful in the light of the theory.

As a side note, this is how the hypothesis was born. In the same device, we have explored the regime of negative S-gate, and found no correlated zero-bias peaks, this was reported in a previous paper (https://www.nature.com/articles/s41567-020-01107-w). Here we focus on positive S-gate. In this regime, we find delocalized states near zero field (up to about 0.5T). However, they are not accompanied by correlated zero-bias peaks either. We understand this based on our model that includes disorder. The disorder is too high to support well-separated Majorana modes; the corresponding wave functions have low characteristic momentum and, consequently, are very susceptible to disorder (which is likely to localize them close to the same end of the system, although some small spatial separation may be present). Only higher momentum states remain delocalized, because their energies are greater than the characteristic disorder energy. This insight came from the model. When we were doing these experiments, we had no benefit of the insight from the model. Starting from the most basic ideas, we reasoned that it will not be possible to see Majorana modes in a regime where all states are localized. Because that would be a low-density regime fully dominated by disorder. We then attempted to find any delocalized states, which we found at higher density. Once we done that, we stayed there to look for zero-bias peaks correlated on both ends, which we could not find. This is how this paper came to be.

First, ideally one should look for Majorana zero modes (MZMs) in a regime where there is a good superconducting gap in the absence of a magnetic field, and hence there should be no delocalized states.

The induced gap here is close to as good as it gets in this materials system, which is NbTiN on InSb. This is one of our best devices. There are subgap states at zero field in all published zero-bias peak experiments. One of the points of our work is to test the Majorana hypothesis deeper in devices close to those that have been used for this purpose.

In general, there is no connection between the presence/absence of delocalized states and the presence/absence of subgap features. On the one hand, there can be subgap features generated by localized states (which is likely the case in all long wires). Also, the presence of delocalized states does not necessarily lead to subgap features (e.g., in a long-enough system the delocalized states will form a quasi-continuum that defines the induced gap edge). In a system that is 400 nm long it is reasonable to expect subgap features at zero field due to finite size effects in the nanowire, even in the absence of strong disorder and localization associated with it. Finally, we emphasize that the delocalized states are not zero-energy states! They generate zero-energy features (even at zero magnetic field) as a result of having finite spectral broadening, which is natural for states coupled to (metallic) leads.

Furthermore, the authors essentially speculated before that the delocalized states have little induced superconductivity - again detrimental for MZMs.

We have removed this speculative discussion from the paper.

  1. On page 4, the authors argue that for localized states strong disorder is necessary. Why is this the case? Isn't this due to the limitations of the model, i.e. the choice of potentials, etc.? I can imagine to get quantum dot like states even in the absence of disorder if potentials are chosen accordingly “Strong disorder” should be understood in this context as representing strong potential inhomogeneity, regardless of its underlying source. A quantum dot automatically generates strong inhomogeneity. The point is that inhomogeneity is required and the characteristic energy scale associated with the inhomogeneity has to be larger than some characteristic energy scale associated with the localized states (e.g., their energy relative to the bottom of the band).

On page 4 the discussion is given for a particular model and this is explicitly stated in the text. “Changing the model parameters and the disorder realizations reveals that features (i) and (iii) are rather generic, while the presence of feature (ii), i.e., the localized states, requires a strong-enough disorder potential. The relative visibility of various features depends on the details of the model.”

We do have another paper where we used a different model in which we do not assume any disorder but still get potential inhomogeneity in realistic device geometries including electrostatics (https://arxiv.org/abs/1902.02772).

More generally, an arbitrarily-long uniform wire will only support delocalized states that extend throughout the whole system. If they are present, the Majorana zero modes will, of course, be localized near the boundaries, but the pair of MZMs can also be viewed as a non-local zero energy fermionic (BdG) state with a separation length comparable to the system size. In the presence of disorder, on the other hand, all states become localized (except at the topological phase boundary). The characteristic length scales of different (localized) states depend on the system parameters (e.g., chemical potential and spin-orbit coupling strength), the characteristic k-vector, and the disorder strength. In a short system, states with large-enough characteristic momentum can have localization lengths comparable to the size of the system; hence, they are effectively delocalized states. Our work demonstrates that one can estimate the threshold between localized and the “effectively delocalized states and extract the corresponding disorder strength. This (indirect) method of estimating the disorder strength is new and can prove extremely useful in monitoring the progress in reducing disorder in hybrid devices (to levels consistent with the emergence of topologically protected Majorana modes). Note that the MZMs should be realized in longer wires, but a short three-terminal device similar to that investigated here (and prepared using the same techniques and conditions) can be used to “measure” the disorder strength.

  1. My biggest concern is the applicability of the numerical simulations to the experiment: The authors argued before that the delocalized states are states separated from the superconductor. However, in the 1D numerical model there is just one mode that is well-proximitized. That is, in my opinion, why the numerical simulations need strong disorder to get agreement with experiment. But this conclusion is weak, having started from orthogonal assumptions!

We have removed from the paper all statements about delocalized states being separated from the superconductor. This was an unnecessary assumption on our part. Rather, the important property of delocalized states is that they have large enough momentum to overcome the disorder (for the given system size). Such delocalized states can derive from a single subband or multiple subbands, depending on the subband occupancy of the system. Nonetheless, any given delocalized state should primarily derive from a single subband (i.e. its spectral weight is dominated by a single subband), except in a very strongly disordered regime where the notion of subband is essentially meaningless. We therefore argue that a 1D model captures the essential physics of the device. To clarify this in the text, we have added two sentences at the bottom of page 2 as follows: Note that the effective one-dimensional model is an appropriate approximation as long as the inter-subband spacing is larger than other relevant energy scales, e.g., the disorder strength. Inclusion of multiple subbands (with relatively small inter-subband gaps) is expected to enhance the disorder effects [20,26]. Also: In this context, we note that in a system with multiple occupied bands, localized and delocalized features can coexist within the same range of gate potentials, Vs, but they will be associated with different subbands. More specifically, … With this resubmission, we have also performed additional simulations and compared experimental and numerical data. We believe we have a better handle on the scale of disorder based on this additional work. It is summarized in the new Figure 6 and the related discussion.

  1. I have a general question: How general are the experimental results for hybrid nanowire systems? In particular, the experiments are performed with sputtered NbTiN which is presumably very disordered. What would you expect to change with e.g. epitaxial superconductors?

Specifically, on the disorder in NbTiN, it is likely that not the atomic-scale disorder in the ternary alloy, but the graininess of the film that is the bigger factor at the moment. From this point of view, the so-called ‘epitaxial’ films are also grainy. In addition, Al thin films have strong surface disorder, which can induce disorder in the proximitized wire. In the most explored ‘epitaxial’ Al devices, no correlated zero-bias states were observed. And only one recent claim by Microsoft exists which some of us find incorrect. One exception is Pb on InAs, which grows with very large grains. But there has been very little work on that system. In terms of the actual Majorana measurements - nothing so far.

---

## Round 2 · Referee Report · Anonymous (Referee 1) · 2023-1-31

Report

The revised manuscript has considerably improved. The Authors have removed some of the more speculative passages, and added a most useful Fig. 6 demonstrating that a quantitatice extraction of disorder strength is possible based on their measurements. Further systematic study is probably needed to assess the reliability and usefulness of this method, e.g. to assess the variance of the results when changing the input data, but that can be left to future work. As it stands, this addition satisfactorily answers one of my main criticisms from the previous report.

I do have one question on this new part. The Authors fix the correlation length of the disorder to 15 nm. They say this is based on the results of Ref. 35, though that reference reports values depending on charge impurity concentration. It would be useful if the Authors could say something about the sensitivity of the results on the correlation length, also because different sources of disorder than charge impurities (e.g. grains in the superconductor) may be characterized by different correlation lengths.

I found that most of the replies to my comments are also satisfactory. In some cases I stand corrected in my remarks, such as on the possible interpretation of the data in terms of Coulomb oscillations. In some other cases I am not fully convinced, but even in these cases I agree with the Authors that the methods and data have been shared to a degree such that readers can form their own judgement. This holds true, in particular, for my concerns related to the superconducting terminal being floating and the other technical details on three-terminal measurements.

Therefore, I don't find it helpful to prolong the debate on lingering disagreements, and based on the improvements I recommend publication of this work in SciPost (pending a possible minor revision on the correlation length based on my question above).

For completeness, let me just answer to two specific points.

First, the Authors state that:

"Bottom-line, identifying starters/nonstarters based on “hardness” of the gap at zero magnetic field is just another misconception among many affecting this field."

I never stated that a hard gap is an indicator of a disorder-free system. I maintain that my statement that a soft gap at B=0 is a "non-starter for a gapped topological phase to occur" is an accurate one. I wish the Authors did not rely on straw man arguments to accuse others of "misconceptions".

Second, the Authors write:

"References are not for giving credit, they are to help readers. We now add those references."

The Authors are free to choose their own deontology. I believe that references in scientific articles are appropriate for both purposes: to give proper credit and/or to help readers. This belief informed my suggested references. I am glad the Authors accepted the suggestions.

---

## Round 2 · Referee Report · Anonymous (Referee 2) · 2023-2-19

Report

The authors replied to my comments and questions in a satisfactory way. Fig. 6 strengthens the message of this work. However, the estimation of disorder is rather speculative and additional measurements and statistics are necessary to draw strong conclusions. Nonetheless, I believe that this work is a good starting point for characterizing disorder in these hybrid structures.

As a minor comment, the resolution of Fig. 5 and Fig. 6 is poor. I would appreciate if the authors could fix this technical issue.

---

## Editorial Decision

published